# Particle Semi-Implicit Variational Inference

**Jen Ning Lim**
University of Warwick
Coventry, United Kingdom
Jen-Ning.Lim@warwick.ac.uk

**Adam M. Johansen**
University of Warwick
Coventry, United Kingdom
a.m.johansen@warwick.ac.uk

## Abstract

Semi-implicit variational inference (SIVI) enriches the expressiveness of variational families by utilizing a kernel and a mixing distribution to hierarchically define the variational distribution. Existing SIVI methods parameterize the mixing distribution using implicit distributions, leading to intractable variational densities. As a result, directly maximizing the evidence lower bound (ELBO) is not possible, so they resort to one of the following: optimizing bounds on the ELBO, employing costly inner-loop Markov chain Monte Carlo runs, or solving minimax objectives. In this paper, we propose a novel method for SIVI called Particle Variational Inference (PVI) which employs empirical measures to approximate the optimal mixing distributions characterized as the minimizer of a free energy functional. PVI arises naturally as a particle approximation of a Euclidean–Wasserstein gradient flow and, unlike prior works, it directly optimizes the ELBO whilst making no parametric assumption about the mixing distribution. Our empirical results demonstrate that PVI performs favourably compared to other SIVI methods across various tasks. Moreover, we provide a theoretical analysis of the behaviour of the gradient flow of a related free energy functional: establishing the existence and uniqueness of solutions as well as propagation of chaos results.

## 1 Introduction

In Bayesian inference, a quantity of vital importance is the posterior $p(x|y) = p(x,y)/\int p(x,y)\mathrm{d}x$, where $p(x,y)$ is a probabilistic model, $y$ denotes the observed data, and $x$ the latent variable. An ever-present issue in Bayesian inference is that the posterior is often intractable. This is because the normalizing constant is available only in the form of an integral, and approximation methods are required. One popular method is variational inference (VI) (Jordan, 1999; Wainwright and Jordan, 2007; Blei et al., 2017). The essence of VI is to approximate the posterior with a member from a variational family $\mathcal{Q}$ where each element of $\mathcal{Q}$ is a distribution $q_\theta$ (called "variational distribution") parameterized by $\theta$. These parameters $\theta$ are obtained via minimizing a distance or discrepancy (or an approximation of it) between the posterior $p(\cdot|y)$ and the variational distribution $q_\theta$.

Here, we focus on semi-implicit variational inference (SIVI) (Yin and Zhou, 2018). It enables a rich variational family by utilizing variational distributions, which we refer to as semi-implicit distributions (SIDs), defined as

$$q_{k,r}(x) := \int k(x|z)r(z)\,\mathrm{d}z, \tag{1}$$

where $k : \mathbb{R}^{d_x} \times \mathbb{R}^{d_z} \to \mathbb{R}_+$ is a kernel satisfying $\int k(x|z)\mathrm{d}x = 1$; $r \in \mathcal{P}(\mathbb{R}^{d_z})$ is the mixing distribution and $\mathcal{P}(\mathbb{R}^{d_z})$ denotes the space of distributions with support $\mathbb{R}^{d_z}$, with its usual Borel $\sigma$-field, with finite second moments. Here, and throughout, we assume that the distributions and kernels of interest admit densities. SIDs are very flexible (Yin and Zhou, 2018) and can express complex properties, such as skewness, multimodality, and kurtosis. These properties might be present

in the posterior but typical variational families may fail to capture them. There are various approaches to parameterizing these variational distributions: current techniques utilize neural networks built on top of existing kernels (e.g., Gaussian kernels) to define more complex kernels (Titsias and Ruiz, 2019), and/or utilize pushforward distributions (a.k.a., implicit distributions (Huszár, 2017)) (Yin and Zhou, 2018). On choosing a parameterization, an approximation to the posterior is obtained by minimizing the exclusive Kullback-Leibler (KL) divergence. This optimization has the same solution as minimizing the free energy (or the negative evidence lower bound) defined as

$$\mathcal{E}(k, r) := \int \log \frac{q_{k,r}(x)}{p(x, y)} q_{k,r}(\mathrm{d}x). \tag{2}$$

However, since the integral in $q_{k,r}$ is typically intractable, directly optimizing $\mathcal{E}$ is not feasible. As a result, SIVI algorithms focus on designing tractable objectives by using upper bounds of $\mathcal{E}$ (Yin and Zhou, 2018); expensive Markov Chain Monte Carlo (MCMC) chains to estimate the gradient of $\mathcal{E}$ (Titsias and Ruiz, 2019); and optimizing different objectives such as score matching which results in min-max objectives (Yu and Zhang, 2023).

In our work, we propose an alternative parameterization for SIDs: kernels are constructed as before (with parameter space denoted by $\Theta$) whereas the mixing distribution $r$ is obtained by optimizing over the whole space $\mathcal{P}(\mathbb{R}^{d_z})$. We motivate the case for minimizing a regularized version of the free energy $\mathcal{E}$ denoted by $\mathcal{E}_\lambda$ (see Eq. (4)); thus, SIVI can be posed as the following optimization problem: $\arg\min_{(\theta, r) \in \Theta \times \mathcal{P}(\mathbb{R}^{d_z})} \mathcal{E}_\lambda(\theta, r)$. As a means to solving the SIVI problem, we construct a gradient flow that minimizes $\mathcal{E}_\lambda$ where the space $\Theta \times \mathcal{P}(\mathbb{R}^{d_z})$ is equipped with the Euclidean–Wasserstein geometry (Jordan et al., 1998; Kuntz et al., 2023). Via discretization, we obtain a practical algorithm for SIVI called *Particle Variational Inference* (PVI) that does not rely upon upper bounds of $\mathcal{E}$, MCMC chains, or solving minimax objectives.

Our main contributions are as follows: (1) we introduce a Euclidean–Wasserstein gradient flow minimizing $\mathcal{E}_\lambda$ as means to perform SIVI; (2) we develop a practical algorithm, PVI, which arises as a discretization of the gradient flow that allows for general mixing distributions; (3) we empirically compare PVI compared with other SIVI approaches across toy and real-world experiments and find that it compares favourably; (4) we study the behaviour of the gradient flow of a related free energy functional to establish existence and uniqueness of solutions (Prop. 8) as well as propagation of chaos results (Prop. 9).

The structure of this paper is as follows: in Section 2, we begin with a discussion of previous approaches to parameterizing SIDs and their relationship with one another. Then, in Section 3, we show how PVI is developed: beginning with designing a well-defined loss functional, the construction of the gradient flow, and how to obtain a practical algorithm. In Section 4, we study properties of a related gradient flow; and, in Section 5, we conclude with experiments to demonstrate the efficacy of our proposal. For sake of brevity, we defer our discussion of related works to App. A.

## 2    On implicit mixing distributions in SIDs

This section outlines existing approaches to parameterizing SIDs with implicit distributions and how these choices affect the resulting variational family. Before we begin, we shall summarize the key assumptions of SIVI. The kernel $k$ is assumed to be a reparametrized distribution in the sense of Salimans and Knowles (2013); Kingma and Welling (2014); Ruiz et al. (2016). In other words, the kernel $k$ is defined by the pair $(\phi, p_k)$ where $\phi : \mathbb{R}^{d_z} \times \mathbb{R}^{d_x} \to \mathbb{R}^{d_x}$ and $p_k \in \mathcal{P}(\mathbb{R}^{d_x})$ such that $k(\cdot|z) = \phi(z, \cdot)_{\#} p_k$ Furthermore, to ensure that it admits a tractable density, the map $\epsilon \mapsto \phi(z, \epsilon)$ is assumed to be a diffeomorphism for all $z \in \mathbb{R}^{d_z}$ with its inverse map written as $\phi^{-1}(z, \cdot)$. From the change-of-variable formula, its density is given as $k(\cdot|z) = p_k(\phi^{-1}(z, \cdot)) |\det \nabla_x \phi^{-1}(z, \cdot)|$. We sometimes write $k_{\phi, p_k}$ to denote the underlying $\phi$ and $p_k$ explicitly. Furthermore, the kernel $k$ is assumed to be computable and differentiable with respect to both arguments.

Several approaches to the parameterization of SIDs have been explored in the literature. One can define the variational family by choosing the kernel and mixing distribution from sets $\mathcal{K}$ and $\mathcal{R}$ respectively, i.e., the variational family is $\mathcal{Q}(\mathcal{K}, \mathcal{R}) := \{q_{k,r} : k \in \mathcal{K}, r \in \mathcal{R}\}$. Yin and Zhou (2018) focused on a fixed kernel $k$ with $r$ being a pushforward (or "implicit") distribution, i.e., $r \in \{g_{\#} p_r : g \in \mathcal{G}\} =: \mathcal{R}_{\mathcal{G}; p_r}$ where $\mathcal{G}$ is a subset of measurable mappings from the sample space of $p_r$ to $\mathbb{R}^{d_z}$. Thus, the $\mathcal{Q}_{\texttt{YiZ}}$-variational family is $\mathcal{Q}(\{k\}, \mathcal{R}_{\mathcal{G}; p_r})$. On the other hand, Titsias and Ruiz

(2019) considered a fixed mixing distribution $r$ with $k$ belonging to some parameterized class $\mathcal{K}$. The typical example is one in which each kernel is defined by composing an existing kernel $k_{\phi,p_k}$ with a function $f \in \mathcal{F}$, the result is $k_{f;\phi,p_k}(\cdot|z) := k_{\phi(f(\cdot),\cdot),p_k}(\cdot|z) = \phi(f(z),\cdot)_\# p_k$ which clearly satisfies the reparameterization assumption. We denoted this kernel class as $\mathcal{K}_{\mathcal{F};\phi,p_k} := \{k_{f;\phi,p_k} : f \in \mathcal{F}\}$ and its respective $\mathcal{Q}_{\mathtt{TR}}$-variational family is $\mathcal{Q}(\mathcal{K}_{\mathcal{G};\phi,p_k}, \{r\})$. In Yu and Zhang (2023), they combine both parameterization for $\mathcal{K}$ and $\mathcal{R}$, i.e., the $\mathcal{Q}_{\mathtt{YuZ}}$-variational family is $\mathcal{Q}(\mathcal{K}_{\mathcal{F};\phi,p_k}, \mathcal{R}_{\mathcal{G};p_r})$. We note that this is how the variational family is presented in Yu and Zhang (2023, see Sec. 2) but the authors used $\mathcal{Q}_{\mathtt{TR}}$-variational family in experiments, i.e., $r$ was fixed. While $\mathcal{Q}_{\mathtt{YuZ}}$ might seem like it defines a larger variational family than the other approaches, under these common parameterization practices, we show that they define the same variational family.

**Proposition 1** ($\mathcal{Q}_{\mathtt{YuZ}} = \mathcal{Q}_{\mathtt{YiZ}} = \mathcal{Q}_{\mathtt{TR}}$). *Given a $\mathcal{Q}_{\mathtt{YuZ}}$-variational family of the form $\mathcal{Q}_{\mathtt{YuZ}} := \mathcal{Q}(\mathcal{K}_{\mathcal{F};\phi,p_k}, \mathcal{R}_{\mathcal{G};p_r})$, then there is a $\mathcal{Q}_{\mathtt{YiZ}}$-variational family and $\mathcal{Q}_{\mathtt{TR}}$-variational family (i.e., $\mathcal{Q}_{\mathtt{TR}} := \mathcal{Q}(\mathcal{K}_{\mathcal{F} \circ \mathcal{G};\phi,p_k}, \{p_r\})$ and $\mathcal{Q}_{\mathtt{YiZ}} := \mathcal{Q}(\{k_{\phi,p_k}\}, \mathcal{R}_{\mathcal{F} \circ \mathcal{G};p_r}))$ such that $\mathcal{Q}_{\mathtt{YuZ}} = \mathcal{Q}_{\mathtt{YiZ}} = \mathcal{Q}_{\mathtt{TR}}$.*

The proof can be found in App. D. This proposition shows that $\mathcal{Q}_{\mathtt{YuZ}}$-parameterization defines the "same" variational family as $\mathcal{Q}_{\mathtt{YiZ}}$ and $\mathcal{Q}_{\mathtt{TR}}$ when we parametrize $\mathcal{R}$ with push-forward distributions. In practice, $\mathcal{F}$ and $\mathcal{G}$ are parameterized by neural networks hence $\mathcal{Q}_{\mathtt{YuZ}}$ can be viewed as $\mathcal{Q}_{\mathtt{YiZ}}$ or $\mathcal{Q}_{\mathtt{TR}}$ with a deeper neural networks $\mathcal{F} \circ \mathcal{G}$. This simplification is a direct result of using push-forward distributions. Although this parametrization has shown promise e.g., Goodfellow et al. (2020)), they have issues with expressivity particularly when distributions are disconnected (Salmona et al., 2022). In our work, we follow in $\mathcal{Q}_{\mathtt{YuZ}}$-variational families, but, we avoid the use of push-forward distributions. Instead, we propose to directly optimize over $\mathcal{P}(\mathbb{R}^{d_z})$ and so, our variational family does not simply reduce to $\mathcal{Q}_{\mathtt{YiZ}}$ or $\mathcal{Q}_{\mathtt{TR}}$.

## 3 Particle Variational Inference

In this section, we present our proposed method for SIVI, called *particle variational inference* (PVI). Similar to prior SIVI methods, the algorithm utilizes kernels (denoted by $k_\theta$) with parameters $\Theta$ which satisfy the assumptions listed in Section 2. One example is $k_\theta \in \mathcal{K}_{\Theta;\phi,p_k}$ where $\Theta$ is a function space induced by a neural network. We slightly abuse the notation $\Theta$ to also indicate its corresponding weight space $\mathbb{R}^{d_\theta}$. The novelty of this algorithm is that, for the mixing distribution, we directly optimize over the space $\mathcal{P}(\mathbb{R}^{d_z})$ which loosens the requirement for the neural network in the kernel to learn complex mappings. The result is a "simpler" optimization procedure and increases expressivity over existing methods. Thus, the variational parameters of PVI are $(\theta, r) \in \Theta \times \mathcal{P}(\mathbb{R}^{d_z}) =: \mathcal{M}$ with its corresponding variational distribution defined as $q_{\theta,r} := \int k_\theta(\cdot|z)r(z)\mathrm{d}z$. PVI arises naturally as a discretization of a gradient flow minimizing a suitably defined free energy on $\Theta \times \mathcal{P}(\mathbb{R}^{d_z})$ endowed with the Euclidean–Wasserstein geometry (Jordan et al., 1998; Ambrosio et al., 2005; Kuntz et al., 2023). In Section 3.1, we begin by constructing a suitably defined free energy functional; then, in Section 3.2, we formulate its gradient flow; finally, in Section 3.3, we construct PVI from its gradient flow.

### 3.1 Free energy functional

As with other VI algorithms, we are interested in finding variational parameters that minimize $(\theta, r) \mapsto \mathsf{KL}(q_{\theta,r}, p(\cdot|y))$. This optimization problem can be cast equivalently as:

$$\operatorname*{arg\,min}_{(\theta,r)\in\mathcal{M}} \mathcal{E}(\theta, r), \quad \text{where } \mathcal{E} : \mathcal{M} \to \mathbb{R} : (\theta, r) \mapsto \int q_{\theta,r}(x) \log \frac{q_{\theta,r}(x)}{p(x,y)} \, \mathrm{d}x. \quad (3)$$

Before we can solve this problem, we must ensure that it is *well-posed*. In other words, it must admit minimizers in $\mathcal{M}$. In the following proposition, we outline various properties of $\mathcal{E}$:

**Proposition 2.** *Assume that the evidence is bounded $\log p(y) < \infty$ and $k$ is bounded; then we have that $\mathcal{E}$ is (i) lower bounded, (ii) lower semi-continuous (l.s.c.), and (iii) non-coercive.*

The proof can be found in App. E.1. Prop. 2 tells us that even though $\mathcal{E}$ possesses many of the properties one looks for in a meaningful minimization functional, it lacks coercivity (in the sense of Dal Maso (2012, Definition 1.12)): a sufficient property to establish the existence of solutions. The key to showing non-coercivity is that we can construct a kernel $k_\theta(x|z)$ that does not depend on $z$.

At first glance, this issue might seem contrived but we note that this problem is closely related to the problem of posterior collapse (Lucas et al., 2019; Wang et al., 2021). To address non-coercivity, we propose to utilize regularization and define the regularized free energy as:

$$\mathcal{E}_\lambda(\theta, r) := \mathbb{E}_{q_{\theta,r}(x)}\left[\log \frac{q_{\theta,r}(x)}{p(x,y)}\right] + \mathsf{R}_\lambda(\theta, r), \tag{4}$$

where $\mathsf{R}_\lambda$ is a regularizer with parameters $\lambda$. In Prop. 3, we show that if $\mathsf{R}_\lambda$ is sufficiently regular, then the $\mathcal{E}_\lambda$ enjoys better properties than its unregularized counterpart $\mathcal{E}$.

**Proposition 3.** *Under the assumptions of Prop. 2, if $\mathsf{R}_\lambda$ is coercive and l.s.c., then the regularized free energy $\mathcal{E}_\lambda$ is (i) lower bounded, (ii) l.s.c., (iii) coercive. Hence it admits at least one minimizer in $\mathcal{M}$.*

The proof can be found in App. E.2. From here forward, we shall focus on regularizers of the form $\mathsf{R}_\lambda^\mathrm{E} : (\theta, r) \mapsto \lambda_r \mathsf{KL}(r, p_0) + \lambda_\theta \mathsf{R}_\theta(\theta)$ where $\lambda = \{\lambda_r, \lambda_\theta\}$ are the regularization parameters and $p_0$ is a predefined reference distribution. As long as $\mathsf{R}_\theta$ is l.s.c., coercive and $\lambda_\theta, \lambda_r > 0$, the resulting regularizer $\mathsf{R}_\lambda^\mathrm{E}$ will also be l.s.c. and coercive. There are many possible choices for $p_0$ and $\mathsf{R}_\theta$. For $p_0$, this regularizes solutions of the gradient flow toward it; as such, in settings where there is some knowledge or preference about $r$ at hand, we can set $p_0$ to reflect that. In our experiments, we utilize $p_0 = \mathcal{N}(0, M)$ where $M$ is a positive definite (p.d.) matrix. As for $\mathsf{R}_\theta$, there are also many choices. In the context of neural networks, one natural choice is Tikhonov regularization $\frac{1}{2}\|\cdot\|^2$, resulting in weight decay (Hanson and Pratt, 1988) for gradient descent (Loshchilov and Hutter, 2019) which is a popular method for regularizing neural networks. In our experiments, we either use Tikhonov regularization or its simple variant $\|\theta\|_M^2 := \langle\theta, M\theta\rangle$.

## 3.2 Gradient flow

To solve the problem in Eq. (3), we construct a gradient flow that minimizes $\mathcal{E}_\lambda$. To this end, we endow the space $\mathcal{M}$ with a suitable notion of gradient $\nabla_\mathcal{M}\mathcal{E}_\lambda(\theta, r) := (\nabla_\theta \mathcal{E}_\lambda, \nabla_r \mathcal{E}_\lambda)$ where $\nabla_\theta$ and $\nabla_r$ denotes the Euclidean gradient and Wasserstein-2 gradient (Jordan et al., 1998), respectively. The latter gradient is given by $\nabla_r \mathcal{E}_\lambda(\theta, r) := -\nabla_z \cdot (r\nabla_z \delta_r \mathcal{E}_\lambda[\theta, r])$, where $\nabla_z \cdot$ denotes the standard divergence operator and $\delta_r$ denotes the first variation which is characterized in the following proposition.

**Proposition 4** (First Variation of $\mathcal{E}_\lambda$ and $\mathsf{R}_\lambda^\mathrm{E}$). *Assume that $\mathbb{E}_{k_\theta(X|\cdot)}\left|\log \frac{q_{\theta,r}(X)}{p(X,y)}\right| < \infty$ for all $(\theta, r) \in \mathcal{M}$; then the first variation of $\mathcal{E}_\lambda$ is $\delta_r \mathcal{E}_\lambda = \delta_r \mathcal{E} + \delta_r \mathsf{R}_\lambda$ where $\delta_r \mathcal{E}[\theta, r](z) = \mathbb{E}_{k_\theta(X|z)}\left[\log \frac{q_{\theta,r}(X)}{p(X,y)}\right]$, and $\delta_r \mathsf{R}_\lambda^\mathrm{E}[\theta, r] = \lambda_r \log r/p_0$.*

The proof can be found in App. E.3. Thus, the (Euclidean–Wasserstein) gradient flow of $\mathcal{E}_\lambda$ is

$$(\dot\theta_t, \dot r_t) = -\nabla_\mathcal{M}\mathcal{E}_\lambda(\theta_t, r_t) \iff \begin{array}{l} \dot\theta_t = -\nabla_\theta \mathcal{E}_\lambda(\theta_t, r_t) \\ \dot r_t = -\nabla_r \mathcal{E}_\lambda(\theta_t, r_t) = \nabla_z \cdot (r_t \nabla_z \delta_r \mathcal{E}_\lambda[\theta_t, r_t]). \end{array} \tag{5}$$

We now establish that the above gradient flow dynamic is contractive and that if a log-Sobolev inequality Eq. (7) holds, one can also establish exponential convergence. The log-Sobolev inequality is closely related to Polyak–Łojasiewicz inequality (or gradient dominance condition) and is commonly assumed in gradient-based systems to obtain convergence (for instance, see Kim et al. (2024)). This is formally stated in the following proposition and proved in App. E.3.

**Proposition 5** (Contracting Gradient Dynamics). *The free energy $\mathcal{E}_\lambda$ along the flow Eq. (5) is non-increasing and satisfies*

$$\frac{\mathrm{d}}{\mathrm{d}t}\mathcal{E}_\lambda(\theta_t, r_t) = -\|\nabla_\mathcal{M}\mathcal{E}_\lambda(\theta_t, r_t)\|^2 \leq 0, \tag{6}$$

*where $\|\nabla_\mathcal{M}\mathcal{E}_\lambda(\theta, r)\|^2 := \|\nabla_\theta \mathcal{E}_\lambda(\theta, r)\|^2 + \|\nabla_z \delta_r \mathcal{E}_\lambda[\theta, r]\|_r^2$. Moreover, if a log-Sobolev Inequality holds for a constant $\tau \in \mathbb{R}_{>0}$, i.e., for all $(\theta, r) \in \mathcal{M}$, we have*

$$\mathcal{E}_\lambda(\theta, r) - \mathcal{E}_\lambda^* \leq \tau\|\nabla_\mathcal{M}\mathcal{E}_\lambda(\theta, r)\|^2, \tag{7}$$

*where $\mathcal{E}_\lambda^* := \inf_{(\theta,r)\in\mathcal{M}} \mathcal{E}_\lambda(\theta, r)$; then we have exponential convergence*

$$\mathcal{E}_\lambda(\theta_t, r_t) - \mathcal{E}_\lambda^* \leq \exp(-t/\tau)(\mathcal{E}_\lambda(\theta_0, r_0) - \mathcal{E}_\lambda^*).$$

Typically direct simulation of the gradient flow Eq. (5) is intractable as the derivative of the first variation of $R^E_\lambda$ involves $\nabla_z \log r_t$; instead, it is useful to identify the gradient flow with a McKean–Vlasov SDE, for which they share the same Fokker–Planck equation. The key distinction is that the SDE can be simulated without access to $\nabla_z \log r_t$. This SDE, which we term the PVI flow, is given by

$$\mathrm{d}\theta_t = -\nabla_\theta \mathcal{E}_\lambda(\theta_t, r_t)\,\mathrm{d}t, \;\; \mathrm{d}Z_t = b(\theta_t, r_t, Z_t)\,\mathrm{d}t + \sqrt{2\lambda_r}\,\mathrm{d}W_t, \tag{8}$$

where $r_t = \mathrm{Law}(Z_t)$, the drift is $b(\theta, r, \cdot) := -\nabla_z \delta_r \mathcal{E}[\theta, r] + \lambda_r \nabla_z \log p_0$ (with the first variation given in Prop. 4) and $W_t$ is a $d_z$-dimensional Wiener process. A connection between the Langevin diffusion, i.e., $\mathrm{d}Z_t = \nabla_z \log p(Z_t, y)\,\mathrm{d}t + \sqrt{2}\mathrm{d}W_t$, and PVI flow can be observed with the fixed kernel $k_\theta(\mathrm{d}x|z) = \delta_z(\mathrm{d}x)$ and $\lambda_r = 0$, namely, they both satisfy the same Fokker–Planck equation.

### 3.3 A practical algorithm

---

**Algorithm 1** Particle Variational Inference (PVI)

---

**Input:** initialization $(\theta_0, \{Z_{0,m}\}_{m=1}^M)$; regularization parameters $\{\lambda_\theta, \lambda_r\}$; step-sizes $h_\theta$ and $h_r$; number of Monte Carlo samples $L$ (for Eqs. (11) and (12)); and preconditioner $\Psi = (\Psi^\theta, \Psi^r)$.

**for** $k = 1$ **to** $K$ **do**
    $r^M_{k-1} \leftarrow \frac{1}{M} \sum_{m=1}^M \delta_{Z_{k-1,m}}$
    $\theta_k \leftarrow \theta_{k-1} - h_\theta \Psi^\theta \widehat{\nabla}_\theta \mathcal{E}_\lambda(\theta_{k-1}, r^M_{k-1})$              ▷ See Eq. (11)
    $\hat{b}_k \leftarrow Z \mapsto -\widehat{\nabla}_z \delta_r \mathcal{E}[\theta_k, r^M_{k-1}](Z) + \lambda_r \nabla_z \log p_0(Z)$      ▷ See Eq. (12)
    **for** $m = 1$ **to** $M$ **do**
        $Z_{k,m} \leftarrow Z_{k-1,m} + h_r \Psi^r \hat{b}_k(Z_{k-1,m}) + \sqrt{\lambda_r h_r \Psi^r} \eta_{k,m}$      ▷ $\eta_{k,m} \sim \mathcal{N}(0, I_{d_z})$
    **end for**
**end for**
**return** $(\theta_K, \{Z_{K,m}\}_{m=1}^M)$

---

To produce a practical algorithm, we are faced with several practical issues. The first issue we tackle is the *computation of gradients* of expectations for which using standard automatic differentiation is insufficient. The second problem is that these gradients are often ill-conditioned and have different scales in each dimension. This is tackled using preconditioning resulting in *adaptive stepsize*. Finally, to produce computationally feasible algorithms, we show how to *discretize* the PVI flow in both *space* and *time*. PVI is summarised in Algorithm 1.

*Computing the gradients.* In the PVI flow, both the drift of the ODE and SDE include a gradient of an expectation with respect to parameters that define the distribution that is being integrated. Specifically, the terms that contain these gradients are $\nabla_\theta \mathcal{E}_\lambda$ and $\nabla_z \delta_r \mathcal{E}_\lambda$. Fortunately, these gradients can be rewritten as an expectation (as described in Prop. 6) for which the parameters being differentiated w.r.t. is only found in the integrand (see derivation in App. G.1).

**Proposition 6.** *If $\phi$ and $k$ are differentiable, then we have*

$$\nabla_\theta \mathcal{E}(\theta, r) = \mathbb{E}_{p_k(\epsilon)r(z)} \left[ (\nabla_\theta \phi_\theta \cdot [s_{\theta,r} - s_p])(z, \epsilon) \right], \tag{9}$$

$$\nabla_z \delta_r \mathcal{E}[\theta, r](z) = \mathbb{E}_{p_k(\epsilon)} \left[ (\nabla_z \phi_\theta \cdot [s_{\theta,r} - s_p])(z, \epsilon) \right], \tag{10}$$

*where $\nabla_\theta \phi \in \mathbb{R}^{d_\theta \times d_x}$ denotes the Jacobian $(\nabla_\theta \phi)_{ij} = \partial_{\theta_i} \phi_j$ (and similarly for $\nabla_z \phi$); scores are $s_{\theta,r}(z, \epsilon) := \nabla_x \log q_{\theta,r}(\phi_\theta(z, \epsilon))$ (and similarly $s_p(z, \epsilon)$) ; and $\cdot$ denotes the usual matrix-vector multiplication in the sense of $M \cdot v : (z, \epsilon) \mapsto M(z, \epsilon)v(z, \epsilon)$.*

From Eqs. (9) and (10), we can produce Monte Carlo estimators for the gradients, i.e.,

$$\widehat{\nabla}_\theta \mathcal{E}(\theta, r) := \frac{1}{L} \sum_{l=1}^L \mathbb{E}_{z \sim r}[(\nabla_z \phi_\theta \cdot [s_{\theta,r} - s_p])(z, \epsilon_l)], \tag{11}$$

$$\widehat{\nabla}_z \delta_r \mathcal{E}[\theta, r] := \frac{1}{L} \sum_{l=1}^L (\nabla_z \phi_\theta \cdot [s_{\theta,r} - s_p])(\cdot, \epsilon_l), \tag{12}$$

where $\{\epsilon_l\}_{l=1}^L \overset{i.i.d.}{\sim} p_k$. This is an instance of a path-wise Monte-Carlo gradient estimator; a performant estimator that has been shown empirically to exhibit lower variance than other standard estimators (Kingma and Welling, 2014; Roeder et al., 2017; Mohamed et al., 2020).

*Adaptive Stepsizes.* One of the complexities of training neural networks is that their gradient is often poorly conditioned. As a result, for certain problems, the gradients computed from Eq. (9) and Eq. (10) can often produce unstable algorithms without careful tuning of the step sizes. When this occurs, we utilize preconditioners (Staib et al., 2019) to avoid this issue. Let $\Psi^\theta : \Theta \mapsto \mathbb{R}^{d_\theta \times d_\theta}$ and $\Psi^r : \mathbb{R}^{d_z} \mapsto \mathbb{R}^{d_z \times d_z}$ be the precondition for components $\theta$ and $r$ respectively, then the resulting preconditioned gradient flow is given by

$$\mathrm{d}\theta_t = -\Psi^\theta \nabla_\theta \mathcal{E}_\lambda(\theta_t, r_t)\, \mathrm{d}t, \ \ \partial_t r_t = \nabla_z \cdot (r_t \Psi^r \nabla_z \delta_r \mathcal{E}_\lambda[\theta_t, r_t]). \tag{13}$$

If $\Psi^\theta$ and $\Psi^r$ are positive definite, then $\mathcal{E}_\lambda(\theta_t, r_t)$ remains non-increasing, i.e., Eq. (6) holds. As before, this Fokker–Planck equation is satisfied by the following Mckean–Vlasov SDE:

$$\mathrm{d}\theta_t = -\Psi^\theta(\theta_t)\nabla_\theta \mathcal{E}_\lambda(\theta_t, r_t)\, \mathrm{d}t, \tag{14}$$

$$\mathrm{d}Z_t = [\Psi^r(Z_t)b(\theta_t, r_t, Z_t) + \nabla_z \cdot \Psi^r(Z_t)]\, \mathrm{d}t + \sqrt{2\lambda\Psi^r(Z_t)}\mathrm{d}W_t, \tag{15}$$

where $(\nabla_z \cdot \Psi^r)_i = \sum_{j=1}^{d_z} \partial_{z_j}[(\Psi^r)_{ij}]$ and $r_t = \mathrm{Law}(Z_t)$. The equivalence between Eq. (13) and Eqs. (14) and (15) is shown in App. G.2. A simple example for the preconditioner allows the $\theta_t$ and $Z_t$ to have different time scales; ultimately, this results in different step sizes. Another more complex example of preconditioner $\Psi^\theta$ is the RMSProp (Tieleman and Hinton, 2012), and $\Psi^r$ we utilize a preconditioner inspired by RMSProp (see App. G.2). As with other related works (e.g., see Li et al. (2016)), we found that the additional term $\nabla_z \cdot \Psi^r$ can be omitted in practice: it has little effect but incurs a large computational cost.

*Discretization in both space and time.* To obtain an actionable algorithm, we need to discretize the PVI flow in both space and time. For the space discretization, we propose to use a particle approximation for $r_t$, i.e., for a set of particles $\{Z_{t,m}\}_{m=1}^M$ with each satisfying $\mathrm{Law}(Z_{t,m}) = r_t$, we utilize the approximation $r_t^M := \frac{1}{M}\sum_{m=1}^M \delta_{Z_{t,m}}$ which converges almost surely to $r_t$ in the weak topology as $M \to \infty$ by the strong law of large numbers and a countable determining class argument (e.g., see Schmon et al. (2020, Theorem 1.1)). This approximation is key to making the intractable tractable, e.g., Eq. (1) is approximated by $q_{\theta,r_t^M} = \frac{1}{M}\sum_{m=1}^M k_\theta(x|Z_{t,m})$. One obtains a particle approximation to the PVI flow from the following ODE–SDE:

$$\mathrm{d}\theta_t^M = -\nabla_\theta \mathcal{E}_\lambda(\theta_t^M, r_t^M)\, \mathrm{d}t, \ \ \forall m \in [M] : \mathrm{d}Z_{t,m}^M = b(\theta_t^M, r_t^M, Z_{t,m}^M)\, \mathrm{d}t + \sqrt{2\lambda_r}\, \mathrm{d}W_{t,m},$$

where $[M] := \{1, \ldots, M\}$. As for the time discretization, we employ Euler–Maruyama discretization with step-size $h$ which (using an appropriately defined preconditioner) can be decoupled into different stepsizes for $\theta_t$ and $Z_t$ denoted by $h_\theta$ and $h_r$ respectively.

## 4 Theoretical analysis

We are interested in the behaviour of the PVI flow (8). However, a key issue in its study is that the drift in PVI flow might lack the necessary continuity properties to analyze using the existing theory. In this section, we instead analyze the related gradient flow of the more regular functional

$$\mathcal{E}_\lambda^\gamma(\theta, r) := \mathcal{E}^\gamma(\theta, r) + \mathsf{R}_\lambda(\theta, r), \quad \text{where } \mathcal{E}^\gamma(\theta, r) = \mathbb{E}_{q_{\theta,r}(x)}\left[\log \frac{q_{\theta,r}(x) + \gamma}{p(x,y)}\right] \tag{16}$$

for $\gamma > 0$. A similar modified flow was also explored in Crucinio et al. (2024) for similar reasons; they found empirically that, at least when using a tamed Euler scheme, setting $\gamma = 0$ did not cause problems in practice. Similarly, our experimental results for PVI found $\gamma = 0$ did not have issues. To provide an additional measure of confidence in the reasonableness of this regularization and of the use of this functional as a proxy for $\mathcal{E}_\lambda$, we establish that the minima of $\mathcal{E}_\lambda^\gamma$ converge to those of $\mathcal{E}_\lambda$ in the $\gamma \to 0$ limit.

**Proposition 7** ($\Gamma$-convergence and convergence of minima). *Under the same assumptions as Prop. 3, we have that $\mathcal{E}_\lambda^\gamma$ $\Gamma$-converges to $\mathcal{E}_\lambda$ as $\gamma \to 0$ (in the sense of Def. B.1). Moreover, we have as an immediate corollary that*

$$\inf_{(\theta,r)\in\mathcal{M}} \mathcal{E}_\lambda(\theta, r) = \lim_{\gamma\to 0} \inf_{(\theta,r)\in\mathcal{M}} \mathcal{E}_\lambda^\gamma(\theta, r).$$

The proof uses techniques from $\Gamma$-convergence theory (introduced by De Giorgi, see e.g. De Giorgi and Franzoni (1975); see Dal Maso (2012) for a good modern introduction) and can be found in App. F.1. The gradient flow of $\mathcal{E}^\gamma$, which we term $\gamma$-PVI, is given by

$$\mathrm{d}\theta_t^\gamma = -\nabla_\theta \mathcal{E}_\lambda^\gamma(\theta_t^\gamma, r_t^\gamma)\,\mathrm{d}t, \quad \mathrm{d}Z_t^\gamma = b^\gamma(\theta_t^\gamma, r_t^\gamma, Z_t^\gamma)\,\mathrm{d}t + \sqrt{2\lambda_r}\,\mathrm{d}W_t, \qquad (17)$$

where $r_t^\gamma = \mathrm{Law}(Z_t^\gamma)$ and $b^\gamma(\theta, r, \cdot) = -\nabla_z \delta_r \mathcal{E}^\gamma[\theta, r] + \lambda_r \nabla_z \log p_0$. The derivation follows similarly to that in Section 3.2, and is omitted for brevity. The use of $\gamma > 0$ is crucial for establishing key regularity conditions in our analysis. We proceed by stating our assumptions.

**Assumption 1** (Regularity of the target $p$, reference distribution $p_0$, and $\mathsf{R}_\theta$)**.** We assume that $\log p(y)$ is *bounded*; and $p$, $p_0$ and $\mathsf{R}_\theta$ have Lipschitz gradients with constants $K_p, K_{p_0}, K_{\mathsf{R}_\theta}$ respectively: there exists some $B_p \in \mathbb{R}_{>0}$ such that $\log p(y) \leq B_p$; and for any given $y$ there exists a $K_p \in \mathbb{R}_{>0}$ such that $\|\nabla_x \log p(x, y) - \nabla_x \log p(x', y)\| \leq K_p \|x - x'\|$ for all $x, x' \in \mathbb{R}^{d_x}$ (similarly for $p_0$ and $\mathsf{R}_\theta$).

**Assumption 2** (Regularity of $k$)**.** We assume that the kernel $k$ and its gradient is *bounded* and has $K_k$-Lipschitz gradient; i.e., there exist constants $B_k, K_k \in \mathbb{R}_{>0}$ such that $|k_\theta(x|z)| + \|\nabla_{(\theta, x, z)} k_\theta(x|z)\| \leq B_k$, and $\|\nabla_x k_\theta(x|z) - \nabla_x k_{\theta'}(x'|z')\| \leq K_k(\|(\theta, x, z) - (\theta', x', z')\|)$ hold for all $\theta, \theta' \in \Theta$, $z, z' \in \mathbb{R}^{d_z}$, and $x, x' \in \mathbb{R}^{d_x}$.

**Assumption 3** (Regularity of $\phi$ and $p_k$.)**.** We assume that $\phi$ has Lipschitz gradient and bounded gradient. In other words, there is $a_\phi \in \mathbb{R}_{\geq 0}, b_\phi \in \mathbb{R}_{>0}$ such that $\phi$ satisfies $\|\nabla_{(\theta, z)} \phi(z, \epsilon) - \nabla_{(\theta, z)} \phi_{\theta'}(z', \epsilon)\|_F \leq (a_\phi \|\epsilon\| + b_\phi)(\|(\theta, z) - (\theta', z')\|)$ and $\|\nabla_{(\theta, z)} \phi_\theta(z, \epsilon)\|_F \leq (a_\phi \|\epsilon\| + b_\phi)$ for all $(\theta, z), (\theta', z') \in \Theta \times \mathbb{R}^{d_z}, \epsilon \in \mathbb{R}^{d_x}$, where $\|\cdot\|_F$ denotes the Frobenius norm. We also assume that $p_k$ has finite second moments.

Assumptions 2 and 3 are intimately connected; under some regularity conditions, one may imply the other but we shall abstain from this digression for the sake of clarity. Assumptions 2 and 3 are quite mild and hold for popular kernels such as $k_\theta(x|z) = \mathcal{N}(x; \mu_\theta(z), \Sigma)$ under some regularity assumptions on $\mu_\theta$ and $\Sigma$ (which we show in App. C). These are key to establishing that the drift in Eq. (17) is Lipschitz continuous (see Prop. 11 in the App.), from which, we establish the existence and uniqueness of the solutions of Eq. (17).

**Proposition 8** (Existence and Uniqueness)**.** *Under Assumptions 1 to 3, if $\gamma > 0$ and $\mathbb{E}_{p_k(\epsilon)} \|s_{\theta, r}^\gamma(z, \epsilon) - s_p(z, \epsilon)\|$ is bounded (with $s_{\theta, r}^\gamma : (z, \epsilon) \mapsto \nabla_x \log(q_{\theta, r} \circ \phi_\theta(z, \epsilon) + \gamma)$); then, given $(\theta_0, r_0) \in \mathcal{M}$, the solutions to Eq. (17) exists and is unique.*

The proof can be found in App. F.2. Under the same assumptions, we can establish an asymptotic propagation of chaos result that justifies the usage of a particle approximation in place of $r_t^\gamma$ in Eq. (17).

**Proposition 9** (Propagation of chaos)**.** *Under the same assumptions as Prop. 8; we have for any fixed $T$:*

$$\lim_{M \to \infty} \mathbb{E} \sup_{t \in [0, T]} \left\|\theta_t^\gamma - \theta_t^{\gamma, M}\right\|^2 + \mathsf{W}_2^2\left((r_t^\gamma)^{\otimes M}, q_t^{\gamma, M}\right) = 0,$$

*where $(r_t^\gamma)^{\otimes M} = \prod_{i=1}^M (r_t^\gamma)$; $q_t^{\gamma, M} = \mathrm{Law}(\{Z_{t, m}^{\gamma, M}\}_{m=1}^M)$; $\theta_t^{\gamma, M}$ and $Z_{t, m}^{\gamma, M}$ are solutions to*

$$\mathrm{d}\theta_t^{\gamma, M} = -\nabla_\theta \mathcal{E}_\lambda^\gamma(\theta_t^{\gamma, M}, r_t^{\gamma, M})\,\mathrm{d}t, \quad \text{where } r_t^{\gamma, M} = \frac{1}{M} \sum_{m=1}^M \delta_{Z_{t, m}^{\gamma, M}}$$

$$\forall m \in [M] : \mathrm{d}Z_{t, m}^{\gamma, M} = b^\gamma(\theta_t^{\gamma, M}, r_t^{\gamma, M}, Z_{t, m}^{\gamma, M})\,\mathrm{d}t + \sqrt{2\lambda_r}\,\mathrm{d}W_{t, m}.$$

The proof can be found in App. F.3. Having established the existence and uniqueness of the $\gamma$-PVI flow, as well as an asymptotic justification for using particles, we now provide a numerical evaluation to demonstrate the efficacy of our proposal.

## 5 Experiments

In this section, we compare PVI against other semi-implicit VI methods. As described in the App. A, these include unbiased semi-implicit variational inference (UVI) of Titsias and Ruiz (2019), semi-implicit variational inference (SVI) of Yin and Zhou (2018), and the score matching approach (SM)

of Yu and Zhang (2023). Through experiments, we show the benefits of optimizing the mixing distribution; we compare the effectiveness of PVI against other SIVI methods on a density estimation problem on toy examples; and, we compare against other SIVI methods on posterior estimation tasks for (Bayesian) logistic regression and (Bayesian) neural network. The details for reproducing experiments as well as computation information can be found in App. H. The code is available at `https://github.com/jenninglim/pvi`.

## 5.1 Impact of the mixing distribution

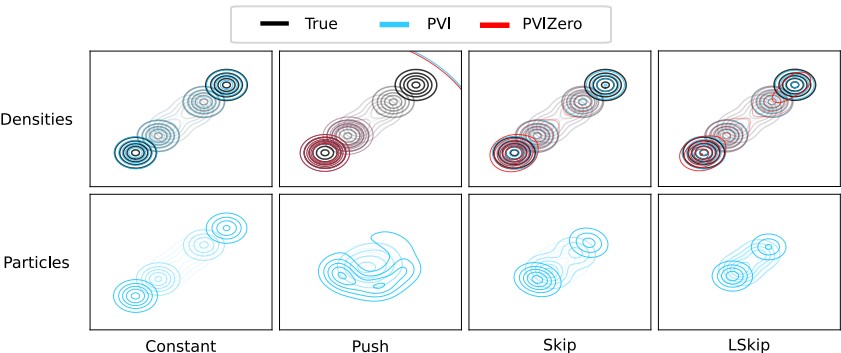

Figure 1: Comparison of PVI and PVIZero on a bimodal mixture of Gaussians for various kernels. The plot shows the density $q_{\theta,r}$ from PVI and PVIZero as well as the KDE plot of $r$ from PVI described by 100 particles.

From Prop. 1, it can be said that ultimately current SIVI methods utilize (directly or indirectly) a fixed mixing distribution whilst PVI does not. We are interested in establishing whether there is any benefit to optimizing the mixing distribution. Intuitively, the mixing distribution can be utilized to express complex properties, such as multimodality, which the neural network kernel $k_\theta$ can then exploit. If the mixing distribution is fixed, this means that the neural network must learn to express these complex properties directly—which can be difficult (Salmona et al., 2022). This intuition turns out to hold, but for the kernel to exploit an expressive mixing distribution, it must be designed well. We illustrate this using the distributions $\frac{1}{2}\mathcal{N}(\mu, I) + \frac{1}{2}\mathcal{N}(-\mu, I)$ for $\mu = \{1, 2, 4\}$ and following kernels: the "Constant" kernel $\mathcal{N}(z, I_2)$; "Push" kernel $\mathcal{N}(f_\theta(z), \sigma_\theta^2 I_2)$; "Skip" kernel $\mathcal{N}(z + f_\theta(z), \sigma_\theta^2 I_2)$; and "LSkip" kernel $\mathcal{N}(Wz + f_\theta(z), \sigma_\theta^2 I_2)$ where $W \in \mathbb{R}^{2 \times 2}$;. We compare the results from PVI and PVIZero (PVI with $h_r = 0$ to result in a fixed $r \approx \mathcal{N}(0, I_2)$) to emulate PVI with a fixed mixing distribution. As $\mu$ gets larger, the complexity of the kernel (or the mixing distribution) must grow to express this (e.g., see (Salmona et al., 2022, Corollary 2)).

Fig. 1 shows the resulting densities and the learnt mixing distribution of PVI and PVIZero for different kernels and various $\mu$. For the constant kernel, PVI can solve this problem by learning a complex mixing distribution to express the multimodality. However, for the push kernel, it can be seen that as $\mu$ gets larger PVI and PVIZero suffer from mode collapse which we suspect is due to the mode-seeking behaviour of using reverse KL and why prior SIVI methods utilized annealing methods (see Yu and Zhang (2023, Section 4.1)). As a remedy, we utilize a Skip kernel which can be seen to improve both PVI and PVIZero. In particular, both PVI and PVIZero were able to successfully express the bimodality in $\mu = 2$; however, PVIZero falls short when $\mu = 4$ while PVI can express the multimodality by learning a bimodal mixing distribution. Since Skip requires $d_z = d_x$, we show that LSkip (which removes the requirement) exhibits a similar behaviour to Skip.

## 5.2 Density estimation

We follow prior works (e.g., Yin and Zhou (2018)) and consider three toy examples whose densities are shown in Fig. 2 (they are given explicitly in App. H.2). In this setting, we use the kernel $k_\theta(x|z) = \mathcal{N}(x; z + f_\theta(z), \sigma_\theta^2 I)$ with $d_z = d_x = 2$ where $f_\theta(z)$ is a neural network whose architecture can be found in App. H.2. As a qualitative measure of performance, Fig. 2 shows the resulting approximating distribution of PVI which can be seen to be a close match to the desired distribution. To compare methods quantitatively, we report the (sliced) Wasserstein distance

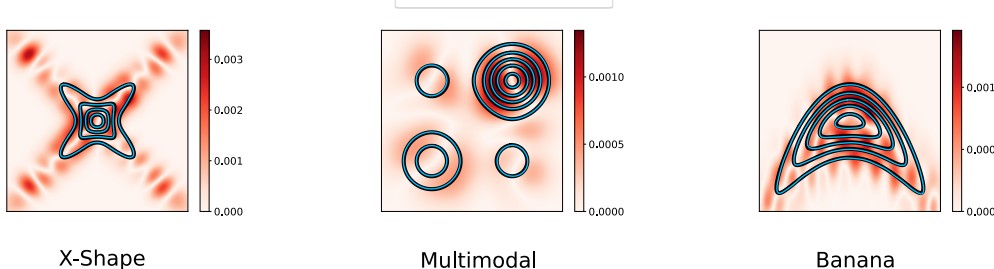

Figure 2: Contour plots of the densities $q_{\theta,r}$ (in blue) against the true densities (in black) for various toy density estimation problems. We also plot the absolute difference in the density of $q_{\theta,r}$ and the true density, i.e., $|q_{\theta,r} - p|$.

(computed by POT (Flamary et al., 2021)) and the rejection power of a state-of-the-art two-sample kernel test (Biggs et al., 2023) between the approximating and true distribution in Table 1. The results reported are the average and standard deviation (from ten independent trials of the respective SIVI algorithms). In each trial, the rejection rate $p$ is computed from 100 tests and the sliced Wasserstein distance is computed from 10000 samples with 100 projections. If the variational approximation matches the distribution, the rejection rate will be at the nominal level of 0.05. It can be seen that PVI consistently performs better than SIVI across all problems. PVI can achieve a rejection rate near nominal levels across all problems whilst other algorithms can achieve good performances on one but not the other. The details regarding how the Wasserstein distance is calculated and the hyperparameters used can be found in App. H.2.

| Problem | PVI | UVI | SVI | SM |
|---|---|---|---|---|
| Banana | $\mathbf{0.06}_{0.02}/\mathbf{0.17}_{0.01}$ | $\mathbf{0.07}_{0.02}/\mathbf{0.11}_{0.03}$ | $0.13_{0.05}/0.31_{0.02}$ | $0.39_{0.24}/0.24_{0.12}$ |
| Multimodal | $\mathbf{0.05}_{0.01}/\mathbf{0.05}_{0.01}$ | $0.65_{0.23}/0.16_{0.07}$ | $0.13_{0.06}/0.08_{0.02}$ | $0.14_{0.05}/0.10_{0.02}$ |
| X-Shape | $\mathbf{0.06}_{0.03}/\mathbf{0.07}_{0.01}$ | $0.23_{0.16}/0.10_{0.04}$ | $0.11_{0.04}/0.12_{0.01}$ | $0.15_{0.11}/0.11_{0.03}$ |

Table 1: This table shows the rejection rate $p$ and average (sliced) Wasserstein distance $w$ for toy density estimation problems. It is written in the format $p/w$ (*lower* is better) with the subscripts showing the standard deviation estimated from 10 independent runs. We indicate in **bold** when the rejection rate minus the standard deviation is lower than the nominal level 0.05, and the algorithm that achieves the lowest Wasserstein score.

## 5.3 Bayesian logistic regression

As with others (Yin and Zhou, 2018), we consider a Bayesian logistic regression problem on the *waveform* dataset (Breiman and Stone, 1984). The model is expressed as $y \mid x, \boldsymbol{w} \sim$ Bernoulli(Sigmoid($\langle x, \overline{\boldsymbol{w}} \rangle$)) with prior $x \sim \mathcal{N}(0, 0.01^{-1} \times I_{22})$ where $(y, \boldsymbol{w}) \in \{0,1\} \times \mathbb{R}^{21}$ is the response and covariates, and $\overline{\boldsymbol{w}} := [1, \boldsymbol{w}]$ is the covariates with appended one for the intercept. The "ground truth" is composed of posterior samples generated from running Markov chain Monte Carlo (MCMC) samples in Yin and Zhou (2018). We use the kernel $k_\theta(x|z) = \mathcal{N}(Wz + f_\theta(z), \exp(\frac{1}{2}[M_\theta + M_\theta^\top]))$ where $\exp$ denotes the matrix exponential which ensures positive definiteness. In Fig. 3, we visually compare certain statistics of the distribution obtained from MCMC and distribution obtained from SIVI methods. Fig. 3a shows the pair-wise marginal posterior distributions for three weights $x_1, x_2, x_3$ chosen at random from MCMC and SIVI approximations; and in Fig. 3b we compare the correlation coefficients obtained from MCMC against SIVI methods. It can be seen that PVI obtains an approximation close to MCMC with most other SIVI methods obtaining similar performance levels (with the exception of SM). See App. H.3 for all the implementation details.

## 5.4 Bayesian neural networks

Following prior works (e.g., Yu and Zhang (2023)), we compare our methods with other baselines on sampling the posterior of the Bayesian neural network for regression problems on a range of real-world datasets. We utilize the LSkip kernel $k_\theta(x|z) = \mathcal{N}(x; Wz + f_\theta(z), \sigma_\theta^2(z)I_{d_x})$. In Table 2,

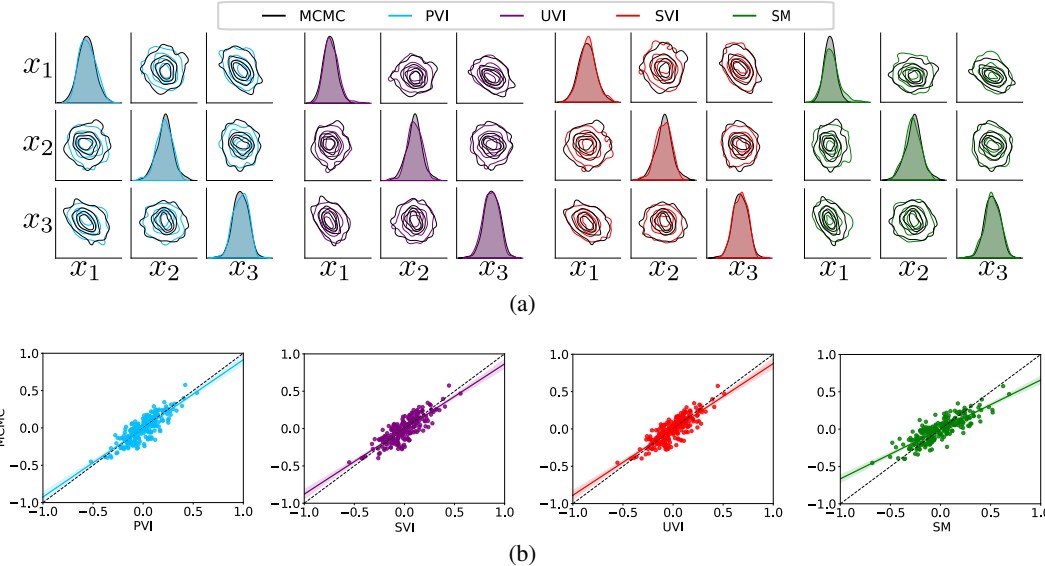

Figure 3: Comparison between SIVI methods and MCMC on Bayesian logistic regression problem. (a) shows the marginal and pairwise approximations of posterior of the weights $x_1, x_2, x_3$, and (b) shows the scatter plot of the correlation coefficient of MCMC ($y$-axis) vs PVI ($x$-axis).

we show the root mean squared error on the test set. It can be seen that PVI performs well, or at least comparable, with other SIVI methods across all datasets. The details regarding the model and other parameters can be found App. H.4.

| Dataset | PVI | UVI | SVI | SM |
|---|---|---|---|---|
| Concrete (Yeh, 2007) | $\mathbf{0.43}_{0.03}$ | $0.50_{0.03}$ | $0.50_{0.04}$ | $0.92_{0.06}$ |
| Protein (Rana, 2013) | $\mathbf{0.87}_{0.05}$ | $0.92_{0.04}$ | $0.92_{0.04}$ | $1.02_{0.03}$ |
| Yacht (Gerritsma et al., 2013) | $\mathbf{0.13}_{0.02}$ | $0.18_{0.02}$ | $0.17_{0.02}$ | $0.98_{0.16}$ |

Table 2: Root mean square error (*lower* is better) for Bayesian neural networks on the test set for various datasets. Here, we write the results in the form $\mu_\sigma$ where $\mu$ is the average RMS and $\sigma$ is its standard error computed over 10 independent trials. We indicate in **bold** the lowest score.

## 6   Conclusion, Limitations, and Future Work

In this work, we frame SIVI as a minimization problem of $\mathcal{E}_\lambda$, and then, as a solution, we study its gradient flow. Through discretization, we propose a novel algorithm called Particle Variational Inference (PVI). Our experiments found that PVI can outperform current SIVI methods. At a marginal increase in computation cost (see App. H) compared with prior methods, PVI can consistently perform better (or at least comparably in the worst cases considered) which we attribute to not imposing a particular form on the mixing distribution. This is a key advantage of PVI compared to prior methods: by not relying upon push-forward mixing distributions and instead using particles, the mixing distribution can express arbitrary distributions when the number of particles is sufficiently large. Furthermore, it is not necessary to tune the family of mixing distributions to obtain good results in particular problems. Theoretically, we study a related gradient flow of $\mathcal{E}_\lambda^\gamma$ and establish desirable properties such as the existence and uniqueness of solutions and propagation of chaos results.

The main limitation of our work is that the theoretical results only apply to the case where $\gamma > 0$; yet, our experiments were performed with $\gamma = 0$ as this is when $\mathcal{E}_\lambda$ corresponds to the (regularized) evidence lower bound. In future work, one can address these limitations by reducing this gap. Furthermore, we found that certain kernels were more amenable than others when exploiting an expressive mixing distribution (e.g., the skip kernel). The question of designing these kernels for PVI (or SIVI more generally) is important for future work.

## Acknowledgments

JNL gratefully acknowledges the funding of the Feuer International Scholarship in Artificial Intelligence. AMJ acknowledges financial support from the United Kingdom Engineering and Physical Sciences Research Council (EPSRC; grants EP/R034710/1 and EP/T004134/1) and by United Kingdom Research and Innovation (UKRI) via grant EP/Y014650/1, as part of the ERC Synergy project OCEAN.

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

# A   Related work

In this section, we outline four areas of related work: semi-implicit variational inference; Euclidean-Wasserstein gradient flows and Wasserstein-gradient flows in VI; mixture models in VI; and finally, the link between SIVI and solving Fredholm equations of the first kind.

At the time of writing, there are three algorithms for SIVI proposed: SVI Yin and Zhou (2018), UVI Titsias and Ruiz (2019), SM Yu and Zhang (2023). Concurrently, Cheng et al. (2024) extended the SM variant by solving the inner minimax objective and simplified the optimization problem. Each had their parameterization of SID (as discussed in Section 2), and their proposed optimization method. SVI relies on optimizing a bound of the ELBO which is asymptotically tight. UVI, like our approach, optimizes the ELBO by using gradients-based approaches. However, one of its terms is the score $\nabla_x \log q_{\theta,r}(x)$ which is intractable. The authors proposed using expensive MCMC chains to estimate it; in contrast to PVI, this term is readily available to us. For SM, they propose to optimize the Fisher divergence, however, to deal with the intractabilities the resulting objective is a minimax optimization problem which is difficult to optimize compared to standard minimization problems.

PVI utilizes the Euclidean–Wasserstein geometry. This geometry and associated gradient flows are initially explored in the context of (marginal) maximum likelihood estimation by Kuntz et al. (2023) and their convergence properties are investigated by Caprio et al. (2024). In Lim et al. (2024), the authors investigated accelerated gradient variants of the aforementioned gradient flow in Euclidean–Wasserstein geometry. The Wasserstein geometry for gradient flows on probability space has received much attention with many works exploring different functionals (for examples, see Arbel et al. (2019); Korba et al. (2021); Li et al. (2023)). In the context of variational inference Lambert et al. (2022) analyzed VI as a Bures–Wasserstein Gradient flow on the space of Gaussian measures.

PVI is reminiscent of mixture distributions which is a consequence of the particle discretization. Mixture models have been studied in prior works as variational distributions (Graves, 2016; Morningstar et al., 2021). In Graves (2016), the authors extended the parameterization trick to mixture distributions; and Morningstar et al. (2021) proposed to utilize mixture models as variational distributions in the framework of Kucukelbir et al. (2017). Although similar, the mixing distribution assists the kernel in expressing complex properties of the true distribution at hand (see Section 5.1) which is an interpretation that mixture distribution lacks.

There is an obvious similarity between SIVI and solving Fredholm equations of the first kind. There is considerable literature on solving such problems; see Crucinio et al. (2024), which is closest in spirit to the approach of the present paper, and references therein. In fact, writing $p(\cdot|y) = \int \tilde{k}(\cdot|z, \theta) r(z) \mathrm{d}z$. with $\tilde{k}(\cdot|z, \theta) \equiv k_\theta(\cdot|z)$ makes the connection more explicit: essentially, one seeks to solve a nonstandard Fredholm equation, with the LHS known only up to a normalizing constant, constraining the solution to be in $\mathcal{P}(\mathcal{Z}) \times \{\delta_\theta : \theta \in \Theta\}$. While Crucinio et al. (2024) develop and analyse a simple Wasserstein gradient flow to address a regularised Fredholm equation, neither the method nor analysis can be applied to the SIVI problem because of this non-trivial constraint. In Yan et al. (2024), the authors also solve a Fredholm-type equation but instead using the Wasserstein–Fisher–Rao geometry (Kondratyev et al., 2016; Gallouët and Monsaingeon, 2017; Chizat et al., 2018; Liero et al., 2018).

# B   $\Gamma$-convergence

The following is one of many essentially equivalent definitions of $\Gamma$-convergence (see Dal Maso (2012); Braides (2002) for comprehensive summaries of $\Gamma$-convergence). We take as definition the following (see Dal Maso (2012, Proposition 8.1), Braides (2002, Definition 1.5)):

**Definition B.1** ($\Gamma$-convergence). Assume that $\mathcal{M}$ is a topological space that satisfies the first axiom of countability. Then a sequence $\mathcal{F}_\gamma : \mathcal{M} \to \mathbb{R}$ is said to $\Gamma$-converge to $\mathcal{F}$ if:

- (lim-inf inequality) for every sequence $(\theta_\gamma, r_\gamma) \in \mathcal{M}$ converging to $(\theta, r) \in \mathcal{M}$, we have
$$\liminf_{\gamma \to 0} \mathcal{F}_\gamma(\theta_\gamma, r_\gamma) \geq \mathcal{F}(\theta, r).$$

- (lim-sup inequality) for any $(\theta, r) \in \mathcal{M}$, there exists a sequence $(\theta_\gamma, r_\gamma) \in \mathcal{M}$, known as a recovery sequence, converging to $(\theta, r)$ which satisfies
$$\limsup_{\gamma \to 0} \mathcal{F}_\gamma(\theta_\gamma, r_\gamma) \leq \mathcal{F}(\theta, r).$$

Γ-convergence corresponds, roughly speaking, to the convergence of the lower semicontinuous enve-lope of a sequence of functionals and, under mild further regularity conditions such as equicoercivity, is sufficient to ensure the convergence of the sets of minimisers of those functionals to the set of minimisers of the limit functional.

## C  On Assumptions 2 and 3

We consider the Gaussian kernel $k_\theta(x|z) = \mathcal{N}(x; \mu_\theta(z), \Sigma)$, i.e.,

$$k_\theta(x|z) = (2\pi)^{-d_x/2}\det(\Sigma)^{-0.5}\exp\left(-\frac{1}{2}(x - \mu_\theta(z))^T\Sigma^{-1}(x - \mu_\theta(z))\right),$$

where $\mu_\theta : \mathbb{R}^{d_z} \mapsto \mathbb{R}^{d_x}$; and $\Sigma \in \mathbb{R}^{d_x \times d_x}$ and is positive definite. In this section, we show that Assumptions 2 and 3 are implied by Assumptions 4 and 5.

**Assumption 4.** $\mu_\theta$ is bounded and $\Sigma$ is positive definite: there exists $B_\mu \in \mathbb{R}_{>0}$ such that the following holds for all $(\theta, z) \in \Theta \times \mathbb{R}^{d_z}$:

$$\|\nabla_{(\theta,z)}\mu_\theta(z)\|_F \le B_\mu,$$

and for any $x \in \mathbb{R}^{d_x} \setminus 0$, $x^T\Sigma x > 0$.

**Assumption 5.** $\mu_\theta$ is Lipschitz and has Lipschitz gradient, i.e., there exist constants $K_\mu \in \mathbb{R}_{>0}$ such that for all $(\theta, z), (\theta', z') \in \Theta \times \mathbb{R}^{d_z}$ the following hold:

$$\|\mu_\theta(z) - \mu_{\theta'}(z')\| \le K_\mu\|(\theta, z) - (\theta', z')\|,$$

$$\|\nabla_{(\theta,z)}\mu_\theta(z) - \nabla_{(\theta,z)}\mu_{\theta'}(z')\|_F \le K_\mu\|(\theta, z) - (\theta', z')\|.$$

### C.1  $k_\theta$ satisfies Assumption 2

In this section, we show that $k_\theta$ satisfies Assumption 2. We first show the boundedness property then the Lipschitz property.

**Boundedness.** First, we shall show that $k_\theta$ is bounded. Clearly, we have $k_\theta(x|z) \in \left[0, (2\pi)^{-d_x/2}\det(\Sigma)^{-0.5}\right]$ hence $|k_\theta|$ is bounded as a consequence of Assumption 4. Now to show that $\|\nabla_{(\theta,x,z)}k_\theta(x|z)\|$ is bounded, we have the following

$$\nabla_x k_\theta(x|z) = -k_\theta(x|z)\Sigma^{-1}(x - \mu_\theta(z)),$$

$$\nabla_z k_\theta(x|z) = \nabla_z\mu_\theta(z)\nabla_\mu\mathcal{N}(x; \mu, \sigma^2 I_{d_x})\mid_{\mu_\theta(z)},$$

$$\nabla_\theta k_\theta(x|z) = \nabla_\theta\mu_\theta(z)\nabla_\mu\mathcal{N}(x; \mu, \sigma^2 I_{d_x})\Big|_{\mu_\theta(z)}.$$

Hence, we have $\|\nabla_{(x,\mu,\sigma)}\mathcal{N}(x; \mu_\theta(z), \sigma^2 I_{d_x})\| < \infty$, from Assumption 4 and using the fact the gradient of a Gaussian density of given covariance w.r.t. $\mu$ is uniformly bounded. Thus, we have shown that $k_\theta$ satisfies the boundedness property in Assumption 2.

**Lipschitz.** For $k_\theta$, one choice of coupling function and noise distribution is $\phi_\theta(z, \epsilon) = \Sigma^{\frac{1}{2}}\epsilon + \mu_\theta(z)$ and $p_k = \mathcal{N}(0, I_{d_x})$ where $\Sigma^{\frac{1}{2}}$ be the unique symmetric and positive definite matrix with $(\Sigma^{\frac{1}{2}})^2 = \Sigma$ (Horn and Johnson, 2012, Theorem 7.2.6); and the inverse map is $\phi_\theta^{-1}(z, x) = \Sigma^{-\frac{1}{2}}(x - \mu_\theta(z))$. Thus, from the change-of-variables formula, we have

$$\begin{aligned}\nabla_x k_\theta(x|z) &= \nabla_x[p_k(\phi_\theta^{-1}(z, x))\det(\nabla_x\phi_\theta^{-1}(z, x))] \\ &= \det(\nabla_x\phi_\theta^{-1}(z, x))\nabla_x[p_k\left(\phi_\theta^{-1}(z, x)\right)] \\ &= \det(\Sigma^{-1/2})\Sigma^{-1/2}\nabla_x p_k(\phi_\theta^{-1}(z, x)) \\ &= \tilde{\Sigma}^{-\frac{1}{2}}\nabla_x p_k(\phi_\theta^{-1}(z, x))\end{aligned}$$

where $\tilde{\Sigma}^{-\frac{1}{2}} := \det(\Sigma^{-1/2})\Sigma^{-1/2}$. Thus, we have

$$\begin{aligned}&\|\nabla_x k_\theta(x|z) - \nabla_x k_{\theta'}(x'|z')\| \\ &\le \|\tilde{\Sigma}^{-\frac{1}{2}}\nabla_x p_k(\phi_\theta^{-1}(z, x)) - \tilde{\Sigma}^{-\frac{1}{2}}\nabla_x p_k(\phi_{\theta'}^{-1}(z', x'))\| \\ &\le \|\tilde{\Sigma}^{-\frac{1}{2}}\|_F\|\nabla_x p_k(\phi_\theta^{-1}(z, x)) - \nabla_x p_k(\phi_{\theta'}^{-1}(z', x'))\| \\ &\le C\|\phi_\theta^{-1}(z, x) - \phi_{\theta'}^{-1}(z', x')\|,\end{aligned}$$

where $C$ is a constant and we use the following facts: $\|\tilde{\Sigma}^{-\frac{1}{2}}\|_F \leq |\det(\Sigma^{-1/2})|\|\Sigma^{-1/2}\|_F < \infty$ following from the fact $\Sigma^{-1/2}$ is positive definite; $p_k$ is a standard Gaussian density function with Lipschitz gradients; and that the inverse map $\phi^{-1}$ is Lipschitz from Assumptions 4 and 5:

$$\|\phi_\theta^{-1}(z, x) - \phi_{\theta'}^{-1}(z', x')\| \leq \|\Sigma^{-\frac{1}{2}}\|_F \|(x, \mu_\theta(z)) - (x', \mu_{\theta'}(z'))\| \leq C' \|(x, \theta, z) - (x', \theta', z')\|.$$

Hence, we have shown that $k_\theta$ satisfies the Lipschitz property of Assumption 2, and so Assumption 2 holds for $k_\theta$.

### C.2 $k_\theta$ satisfies Assumption 3

One can compute the gradient as

$$\nabla_{(\theta, z)} \phi_\theta(z, \epsilon) = \nabla_{(\theta, z)} \mu_\theta(z),$$

and hence $\|\nabla_{(\theta, z)} \phi_\theta(z, \epsilon)\|_F$ is bounded from Assumption 4. The Lipschitz gradient property is immediate from Assumption 5.

$p_k$ has finite second moments since it is a Gaussian with positive definite covariance matrix.

## D  Proofs in Section 2

*Proof of Prop. 1.* We start by showing $\mathcal{Q}_{\text{YuZ}} = \mathcal{Q}_{\text{TR}}$. To this end, we begin by showing the inclusion $\mathcal{Q}_{\text{YuZ}} \subseteq \mathcal{Q}_{\text{TR}}$, i.e., $\mathcal{Q}(\mathcal{K}_{\mathcal{F};\phi,p_k}, \mathcal{R}_{\mathcal{G},p_r}) \subseteq \mathcal{Q}(\mathcal{K}_{\mathcal{F}\circ\mathcal{G};\phi,p_k}, \{p_r\})$. Let $q \in \mathcal{Q}(\mathcal{K}_{\mathcal{F};\phi,p_k}, \mathcal{R}_{\mathcal{G},p_r})$, then there is some $f \in \mathcal{F}$ and $g \in \mathcal{G}$ such that $q = q_{k_{f;\phi,p_k}, g_\# p_r}$. From straight-forward computation, we have

$$q_{k_{f;\phi,p_k}, g_\# p_r} = \mathbb{E}_{z \sim g_\# p_r}[k_{f;\phi,p_k}(\cdot|z)] \stackrel{(a)}{=} \mathbb{E}_{z \sim p_r}[k_{f;\phi,p_k}(\cdot|g(z))] \in \mathcal{Q}(\mathcal{K}_{\mathcal{F}\circ\mathcal{G};\phi,p_k}, \{p_r\}),$$

where (a) follows the law of the unconscious statistician, and the last element-of follows from the fact that $k_{f;\phi,p_k}(\cdot|g(\epsilon))) = \phi(f \circ g(\epsilon), \cdot)_\# p_k \in \mathcal{K}_{\mathcal{F}\circ\mathcal{G};\phi,p_k}$. We can follow the argument above in reverse to obtain the reverse inclusion. Hence, we have obtained as desired.

That $\mathcal{Q}_{\text{YuZ}} = \mathcal{Q}_{\text{YiZ}}$, follows in a similar manner, which we shall outline for completeness: let $q \in \mathcal{Q}(\mathcal{K}_{\mathcal{F};\phi,p_k}, \mathcal{R}_{\mathcal{G},p_r})$, then

$$q = q_{k_{f;\phi,p_k}, g_\# p_r} = \mathbb{E}_{z \sim g_\# p_r}[k_{f;\phi,p_k}(\cdot|z)] = \mathbb{E}_{z \sim f \circ g_\# p_r}[k_{\phi,p_k}(\cdot|z)] \in \mathcal{Q}_{\text{YiZ}}.$$

One can conclude by applying the same logic in the reverse direction. $\square$

## E  Proofs in Section 3

### E.1  Proof of Prop. 2

*Proof of Prop. 2.* ($\mathcal{E}$ is lower bounded). Clearly, we have

$$\mathcal{E}(\theta, r) = \mathsf{KL}(q_{\theta,r}, p(\cdot|y)) - \log p(y) \geq -\log p(y),$$

Hence, we have $\mathcal{E}(\theta, r) \in [-\log p(y), \infty)$ which is lower bounded by our assumption.

($\mathcal{E}$ is lower semi-continuous). Let $(\theta_n, r_n)_{n \in \mathbb{N}}$ be such that $\lim_{n \to \infty} r_n = r$ and $\lim_{n \to \infty} \theta_n = \theta$.

We can split the domain of integration, and write $\mathcal{E}$ equivalently as

$$\mathcal{E}(\theta, r) = \int \underbrace{\mathbb{1}_{[1,\infty)} \left( \frac{p(x,y)}{q_{\theta,r}(x)} \right) \log \left( \frac{q_{\theta,r}(x)}{p(x,y)} \right)}_{\leq 0} q_{\theta,r}(x)\, \mathrm{d}x \tag{18}$$

$$+ \int \underbrace{\mathbb{1}_{[0,1)} \left( \frac{p(x,y)}{q_{\theta,r}(x)} \right) \log \left( \frac{q_{\theta,r}(x)}{p(x,y)} \right)}_{\geq 0} q_{\theta,r}(x)\, \mathrm{d}x \tag{19}$$

We shall focus on the RHS of (18).

Note that we have the following bound

$$
\left| -\mathbb{1}_{[1,\infty)}\left(\frac{p(x,y)}{q_{\theta_n,r_n}(x)}\right) \log\left(\frac{q_{\theta_n,r_n}(x)}{p(x,y)}\right) q_{\theta_n,r_n}(x) \right|
$$

$$
\leq \max\left\{0, \log\left(\frac{p(x,y)}{q_{\theta_n,r_n}(x)}\right) q_{\theta_n,r_n}(x)\right\} \leq \max\{0, C\},
$$

where $C$ is some constant. The last inequality follows from the fact that the evidence is bounded from above and the kernel is bounded. We can apply Reverse Fatou's Lemma to obtain

$$
\limsup_{n\to\infty} -\int \mathbb{1}_{[1,\infty)}\left(\frac{p(x,y)}{q_{\theta_n,r_n}(x)}\right) \log\left(\frac{q_{\theta_n,r_n}(x)}{p(x,y)}\right) q_{\theta_n,r_n}(x)\,\mathrm{d}x
$$

$$
\leq \int \limsup_{n\to\infty}\left(-\mathbb{1}_{[1,\infty)}\left(\frac{p(x,y)}{q_{\theta_n,r_n}(x)}\right) \log\left(\frac{q_{\theta_n,r_n}(x)}{p(x,y)}\right) q_{\theta_n,r_n}(x)\right)\,\mathrm{d}x.
$$

Since we have the following relationships

$$
\limsup_{n\to\infty} \mathbb{1}_{[1,\infty)}\left(\frac{p(x,y)}{q_{\theta_n,r_n}(x)}\right) \leq \mathbb{1}_{[1,\infty)}\left(\frac{p(x,y)}{q_{\theta,r}(x)}\right),
$$

$$
\lim_{n\to\infty} -\log\left(\frac{p(x,y)}{q_{\theta_n,r_n}(x)}\right) = -\log\left(\frac{p(x,y)}{q_{\theta,r}(x)}\right),
$$

$$
\lim_{n\to\infty} q_{\theta_n,r_n} = q_{\theta,r} \text{ pointwise,}
$$

where the first line is from u.s.c. of $\mathbb{1}_{[1,\infty)}$; the second line from the continuity of $\log$; the final line follows from the bounded kernel $k$ assumption and dominated convergence theorem.

Thus, we have that

$$
\limsup_{n\to\infty} -\int \mathbb{1}_{[1,\infty)}\left(\frac{p(x,y)}{q_{\theta_n,r_n}(x)}\right) \log\left(\frac{q_{\theta_n,r_n}(x)}{p(x,y)}\right) q_{\theta_n,r_n}(x)\,\mathrm{d}x
$$

$$
\leq -\int \mathbb{1}_{[1,\infty)}\left(\frac{p(x,y)}{q_{\theta,r}(x)}\right) \log\left(\frac{q_{\theta,r}(x)}{p(x,y)}\right) q_{\theta,r}(x)\,\mathrm{d}x,
$$

Using the fact that $\limsup_{n\to\infty} -x_n = -\liminf_{n\to\infty} x_n$, we have shown that

$$
-\liminf_{n\to\infty} \int \mathbb{1}_{[1,\infty)}\left(\frac{p(x,y)}{q_{\theta_n,r_n}(x)}\right) \log\left(\frac{q_{\theta_n,r_n}(x)}{p(x,y)}\right) q_{\theta_n,r_n}(x)\,\mathrm{d}x
$$

$$
\leq -\int \mathbb{1}_{[1,\infty)}\left(\frac{p(x,y)}{q_{\theta,r}(x)}\right) \log\left(\frac{q_{\theta,r}(x)}{p(x,y)}\right) q_{\theta,r}(x)\,\mathrm{d}x. \tag{20}
$$

Similarly, for the RHS of (19), using Fatou's Lemma (with varying measure and the set-wise convergence of $q_{\theta_n,r_n}$) and using the l.s.c. of $\mathbb{1}_{[0,1)}$, we obtain that

$$
\liminf_{n\to\infty} \int \mathbb{1}_{[0,1)}\left(\frac{p(x,y)}{q_{\theta_n,r_n}(x)}\right) \log\left(\frac{q_{\theta_n,r_n}(x)}{p(x,y)}\right) q_{\theta_n,r_n}(x)\,\mathrm{d}x
$$

$$
\geq \int \mathbb{1}_{[0,1)}\left(\frac{p(x,y)}{q_{\theta,r}(x)}\right) \log\left(\frac{q_{\theta,r}(x)}{p(x,y)}\right) q_{\theta,r}(x)\,\mathrm{d}x. \tag{21}
$$

Hence, combining the bounds (20) and (21), we have that shown that

$$
\liminf_{n\to\infty} \mathcal{E}(\theta_n, r_n) = \liminf_{n\to\infty} \int \log\left(\frac{q_{\theta_n,r_n}(x)}{p(x,y)}\right) q_{\theta_n,r_n}(x)\,\mathrm{d}x
$$

$$
\geq \int \log\left(\frac{q_{\theta,r}(x)}{p(x,y)}\right) q_{\theta,r}(x)\,\mathrm{d}x \geq \mathcal{E}(\theta, r).
$$

In other words, $\mathcal{E}$ is lower semi-continuous.

(Non-Coercivity) To show non-coercivity, we will show that there exists some level set $\{(\theta, r) : \mathcal{E}(\theta, r) \leq \beta\}$ that is not compact. We do this by finding a sequence contained in the level set that does not contain a (weakly) converging subsequence.

Consider the sequence $\Pi := (\theta_n, r_n)_{n \in \mathbb{N}}$ where $\theta_n = \theta_0$; $\|\theta_0\| < \infty$; $r_n = \delta_n$; $k_\theta(x|z) = \mathcal{N}(x; \theta, I_{d_x})$; and $p(x|y) = \mathcal{N}(x; 0, I_{d_x})$. Clearly, we have $q_{\theta,r}(x) = \mathcal{N}(x; \theta, I_{d_x})$ and so $\mathsf{KL}(q_{\theta,r}, p(\cdot|y)) = \frac{1}{2}\|\theta\|^2$. Hence, there is a $\beta < \infty$ such that

$$\mathcal{E}(\theta_n, r_n) = \mathsf{KL}(q_{\theta_n, r_n}, p(\cdot|y)) - \log p(y) \leq \frac{1}{2}\|\theta_0\|^2 - \log p(y) \leq \beta.$$

Thus, we have shown that $\Pi \subset \{(\theta, r) : \mathcal{E}(\theta, r) \leq \beta\}$. However, since the support of the elements of $\{r_n \in \mathcal{P}(\mathbb{R}^{d_z})\}_n$ eventually lies outside a ball of radius $R$ for any $R < \infty$ and hence of any compact set, $\Pi$ is not tight. Hence, Prokhorov's theorem (Shiryaev, 1996, p. 318) tells us that, as $\Pi$ is not tight, it is not relatively compact. We conclude that, as the level set is not relatively compact, the functional is not-coercive. $\qquad\square$

## E.2 Proof of Prop. 3

*Proof of Prop. 3.* (Coercivity) Consider the level set $\{(\theta, r) : \mathcal{E}_\lambda(\theta, r) \leq \beta\}$, which is contained in a relatively compact set. To see this, first note that

$$\{(\theta, r) : \mathcal{E}_\lambda(\theta, r) \leq \beta\} \subseteq \{(\theta, r) : -\log p(y) + \mathsf{R}_\lambda(\theta, r) \leq \beta\}$$
$$\subseteq \{(\theta, r) : \mathsf{R}_\lambda(\theta, r) \leq \beta + \log p(y)\}$$

By coercivity of $\mathsf{R}_\lambda$, i.e., the above level set is relatively compact hence $\mathcal{E}_\lambda$ is coercive.

(Lower semi-continuity) Lower semi-continuity (l.s.c.) follows immediately from the l.s.c. of $\mathcal{E}$ and $\mathsf{R}_\lambda$.

(Existence of a minimizer) The existence of a minimizer follows from Dal Maso (2012, Theorem 1.15) utilizing coercivity and l.s.c. of $\mathcal{E}_\lambda$. $\qquad\square$

## E.3 Proof of Prop. 4

Recall from Santambrogio (2015, Definition 7.12),

**Definition E.1** (First Variation). If $p$ is regular for $F$, the first variation of $F : \mathcal{P}(\mathbb{R}^{d_z}) \to \mathbb{R}$, if it exists, is the element that satisfies

$$\lim_{\epsilon \to 0} \frac{F(p + \epsilon\chi) - F(p)}{\epsilon} = \int \delta_r F[r](z)\chi(\mathrm{d}z),$$

for any perturbation $\chi = \tilde{p} - p$ with $\tilde{p} \in \mathcal{P}(\mathbb{R}^{d_z}) \cap L_c^\infty(\mathbb{R}^{d_z})$ (see Santambrogio (2015, Notation)).

One can decompose the first variation of $\mathcal{E}_\lambda^\gamma$ as:

$$\delta_r \mathcal{E}_\lambda^\gamma[\theta, r] = \delta_r \mathcal{E}^\gamma[\theta, r] + \delta_r \mathsf{R}_\lambda^{\mathrm{E}}[\theta, r].$$

where $\mathcal{E}^\gamma : (\theta, r) \mapsto \int \log\left(\frac{q_{\theta,r}(x) + \gamma}{p(x,y)}\right) q_{\theta,r}(\mathrm{d}x)$. Since $\delta_r \mathsf{R}_\lambda^{\mathrm{E}}[\theta, r] = \lambda_r \delta_r \mathsf{KL}(r|p_0)$, its first variation follows immediately from standard calculations (Ambrosio et al., 2005; Santambrogio, 2015). As for $\delta_r \mathcal{E}^\gamma$, we have the following proposition:

**Proposition 10** (First Variation of $\mathcal{E}^\gamma$). *Assume that for all $(\theta, r, z) \in \mathcal{M} \times \mathbb{R}^{d_z}$,*

$$\mathbb{E}_{k_\theta(X|z)}\left|\log\left(\frac{q_{\theta,r}(X) + \gamma}{p(X, y)}\right)\right| < \infty,$$

*then we obtain*

$$\delta_r \mathcal{E}^\gamma[\theta, r](z) = \mathbb{E}_{k_\theta(X|z)}\left[\log\left(\frac{q_{\theta,r}(X) + \gamma}{p(X, y)}\right) + \frac{q_{\theta,r}(X)}{q_{\theta,r}(X) + \gamma}\right].$$

*Proof.* Since $q_{\theta, r+\epsilon\chi} = \int k_\theta(\cdot|z)(r + \epsilon\chi)(z)\,\mathrm{d}z = q_{\theta,r} + \epsilon q_{\theta,\chi}$, we have

$$\mathcal{E}^\gamma(\theta, r + \epsilon\chi) = \int_{\mathcal{X}} q_{\theta, r+\epsilon\chi}(x) \log\left(\frac{q_{\theta, r+\epsilon\chi}(x) + \gamma}{p(y, x)}\right)\,\mathrm{d}x$$

$$= \int_{\mathcal{X}} [q_{\theta,r} + \epsilon q_{\theta,\chi}](x) \log([q_{\theta,r} + \epsilon q_{\theta,\chi}](x) + \gamma)\,\mathrm{d}x$$

$$- \int_{\mathcal{X}} [q_{\theta,r} + \epsilon q_{\theta,\chi}](x) \log p(y, x)\,\mathrm{d}x.$$

Applying Taylor's expansion, we obtain $(x + \epsilon y)\log(x + \epsilon y + \gamma) = x\log(x + \gamma) + \epsilon y\left(\log(x + \gamma) + \frac{x}{x+\gamma}\right) + o(\epsilon)$, we obtain

$$\mathcal{E}^\gamma(\theta, r + \epsilon\chi) = \int_\mathcal{X} q_{\theta,r}(x) \log \frac{q_{\theta,r}(x) + \gamma}{p(y,x)} \, \mathrm{d}x$$
$$+ \epsilon \int_\mathcal{X} q_{\theta,\chi}(x) \left[\log\left(\frac{q_{\theta,r}(x) + \gamma}{p(y,x)}\right) + \frac{q_{\theta,r}(x)}{q_{\theta,r}(x) + \gamma}\right] \mathrm{d}x + o(\epsilon).$$

Hence, we obtain

$$\lim_{\epsilon \to} \frac{\mathcal{E}^\gamma(\theta, r + \epsilon\chi) - \mathcal{E}^\gamma(\theta, r)}{\epsilon} = \int_\mathcal{X} q_{\theta,\chi}(x) \left[\log \frac{q_{\theta,r}(x) + \gamma}{p(y,x)} + \frac{q_{\theta,r}(x)}{q_{\theta,r}(x) + \gamma}\right] \mathrm{d}x$$

$$= \int_\mathcal{X} \left[\int_\mathcal{Z} k_\theta(x|z)\chi(\mathrm{d}z)\right] \left[\log\left(\frac{q_{\theta,r}(x) + \gamma}{p(y,x)}\right) + \frac{q_{\theta,r}(x)}{q_{\theta,r}(x) + \gamma}\right] \mathrm{d}x$$

$$\stackrel{(a)}{=} \int_\mathcal{Z} \left(\int_\mathcal{X} k_\theta(x|z) \left[\log\left(\frac{q_{\theta,r}(x) + \gamma}{p(y,x)}\right) + \frac{q_{\theta,r}(x)}{q_{\theta,r}(x) + \gamma}\right] \mathrm{d}x\right) \chi(z) \, \mathrm{d}z.$$

One can then identify the desired result. In (a), we appeal to Fubini's theorem for the interchange of integrals whose conditions

$$\int_\mathcal{Z} \int_\mathcal{X} \left| k_\theta(x|z) \left[\log\left(\frac{q_{\theta,r}(x) + \gamma}{p(y,x)}\right) + \frac{q_{\theta,r}(x)}{q_{\theta,r}(x) + \gamma}\right] \chi(z) \right| \mathrm{d}x\mathrm{d}z < \infty, \tag{22}$$

are satisfied by our assumptions. This can be seen from

$$\text{LHS Eq. (22)} \leq \int_\mathcal{Z} \mathbb{E}_{k_\theta(X|z)} \left| \log\left(\frac{q_{\theta,r}(X) + \gamma}{p(X,y)}\right) + \frac{q_{\theta,r}(X)}{q_{\theta,r}(X) + \gamma} \right| |\chi(z)| \mathrm{d}z \leq 0,$$

where we use our assumption and the fact that $\chi$ is absolutely integrable. $\qquad\square$

*Proof of Prop. 5.* The result can be obtained from direct computation. We begin

$$\frac{\mathrm{d}}{\mathrm{d}t} \mathcal{E}_\lambda(\theta_t, r_t) = \left\langle \nabla_\theta \mathcal{E}_\lambda(\theta_t, r_t), \dot\theta_t \right\rangle + \int \delta_r \mathcal{E}_\lambda[\theta_t, r_t] \, \partial_t r_t \, \mathrm{d}z$$

The second term can be simplified

$$\int \delta_r \mathcal{E}_\lambda[\theta_t, r_t] \, \partial_t r_t \, \mathrm{d}z = \int \delta_r \mathcal{E}_\lambda[\theta_t, r_t](z) \nabla_z \cdot (r_t(z)\nabla \delta_r \mathcal{E}_\lambda[\theta_t, r_t](z)) \, \mathrm{d}z$$

$$= -\int r_t \|\nabla_z \delta_r \mathcal{E}_\lambda[\theta_t, r_t](z)\|^2 \, \mathrm{d}z$$

where the last inequality follows from integration by parts. Hence, the claim holds. If the log-Sobolev inequality holds, then we have

$$\frac{\mathrm{d}}{\mathrm{d}t} \left[\mathcal{E}_\lambda(\theta_t, r_t) - \mathcal{E}_\lambda^*\right] = -\|\nabla_\mathcal{M} \mathcal{E}_\lambda[\theta_t, r_t]\| \leq -\frac{1}{\tau}\left[\mathcal{E}_\lambda(\theta_t, r_t) - \mathcal{E}_\lambda^*\right].$$

From Grönwalls inequality, we obtain the desired result. $\qquad\square$

# F  Proofs in Section 4

## F.1  Proof of Prop. 7

*Proof.* We first begin by proving $\Gamma$-convergence directly via its definition, i.e., demonstrating that the liminf inequality holds and establishing the existence of a recovery sequence. The latter follows from pointwise convergence:

$$\lim_{\gamma \to 0} \mathcal{E}_\lambda^\gamma(\theta, r) = \mathcal{E}_\lambda(\theta, r),$$

upon taking $(\theta_\gamma, r_\gamma) = (\theta, r)$ for all $\gamma$.

The liminf inequality can be seen to follow similarly from the l.s.c. argument in App. E.1.

To arrive at the convergence of minima, we invoke Dal Maso (2012, Theorem 7.8) by using the fact that $\mathcal{E}_\lambda^\gamma$ is equi-coercive in the sense of Dal Maso (2012, Definition 7.6). To see that $\mathcal{E}_\lambda^\gamma$ is equi-coercive, note that we have $\mathcal{E}_\lambda^\gamma \geq \mathcal{E}_\lambda$ and $\mathcal{E}_\lambda$ is l.s.c. (from Prop. 3), then applying Dal Maso (2012, Proposition 7.7). $\qquad\square$

## F.2 Proof of Prop. 8

*Proof of Prop. 8.* We can equivalently write the $\gamma$-PVI flow in Eq. (17) as follows

$$\mathrm{d}(\theta_t, Z_t) = \tilde{b}^\gamma(\theta_t, \mathrm{Law}(Z_t), Z_t)\,\mathrm{d}t + \sigma\,\mathrm{d}W_t, \tag{23}$$

where $\sigma = \begin{bmatrix} 0 & 0 \\ 0 & \sqrt{2\lambda_r}I_{d_z} \end{bmatrix}$, and

$$\tilde{b}^\gamma : \mathbb{R}^{d_\theta} \times \mathcal{P}(\mathcal{Z}) \times \mathbb{R}^{d_z} \to \mathbb{R}^{d_\theta + d_z} : (\theta, r, Z) \mapsto \begin{bmatrix} -\nabla_\theta \mathcal{E}_\lambda^\gamma(\theta, r) \\ b^\gamma(\theta, r, Z) \end{bmatrix}.$$

In App. F.4, we show that under our assumptions the drift $\tilde{b}^\gamma$ is Lipschitz. And under Lipschitz regularity conditions, the proof follows similarly to Lim et al. (2024) which we shall outline for completeness.

We begin endowing the space $\Theta \times \mathcal{P}(\mathbb{R}^{d_z})$ with the metric

$$\mathsf{d}((\theta, r), (\theta', r')) = \sqrt{\|\theta - \theta'\|^2 + \mathsf{W}_2^2(q, q')}.$$

Let $\Upsilon \in C([0, T], \Theta \times \mathcal{P}(\mathbb{R}^{d_z}))$ and denote $\Upsilon_t = (\vartheta_t^\Upsilon, \nu_t^\Upsilon)$ for it's respective components. Consider the process that substitutes $\Upsilon$ into (23), in place of the $\mathrm{Law}(Z_t)$ and $\theta_t$,

$$\mathrm{d}(\theta_t^\Upsilon, Z_t^\Upsilon) = \tilde{b}^\gamma(\vartheta_t^\Upsilon, \nu_t^\Upsilon, Z_t^\Upsilon)\,\mathrm{d}t + \sigma\,\mathrm{d}W_t.$$

whose existence and uniqueness of strong solutions are given by Carmona (2016)[Thereom 1.2].

Define the operator

$$F_T : C([0, T], \Theta \times \mathcal{P}(\mathbb{R}^{d_z})) \to C([0, T], \Theta \times \mathcal{P}(\mathbb{R}^{d_z})) : \Upsilon \to (t \mapsto (\theta_t^\Upsilon, \mathrm{Law}(Z_t^\Upsilon)).$$

Let $(\theta_t, Z_t)$ denote a process that is a solution to (23) then the function $t \mapsto (\theta_t, \mathrm{Law}(Z_t))$ is a fixed point of the operator $F_T$. The converse also holds. Thus, it is sufficient to establish the existence and uniqueness of the fixed point of the operator $F_T$. For $\Upsilon = (\vartheta, \nu)$ and $\Upsilon' = (\vartheta', \nu')$

$$\begin{aligned}
\|\theta_t^\Upsilon - \theta_t^{\Upsilon'}\|^2 + \mathbb{E}[\|Z_t^\Upsilon - Z_t^{\Upsilon'}\|]^2 &= \mathbb{E}\left\|\int_0^t \tilde{b}^\gamma(\vartheta_s, \nu_s, Z_s^\Upsilon) - \tilde{b}^\gamma(\vartheta_s', \nu_s', Z_s^{\Upsilon'})\,\mathrm{d}s\right\|^2 \\
&\leq tC \int_0^t \left[\mathbb{E}\|Z_s^\Upsilon - Z_s^{\Upsilon'}\|^2 + \|\vartheta_s - \vartheta_s'\|^2 + \mathsf{W}_1^2(\nu_s, \nu_s')\right]\,\mathrm{d}s \\
&\leq C(t) \int_0^t [\mathsf{W}_2^2(\nu_s, \nu_s') + \|\vartheta_s - \vartheta_s'\|^2]\,\mathrm{d}s,
\end{aligned}$$

where we apply Jensen's inequality; $C_r$-inequality; Lipschitz drift of $\tilde{b}^\gamma$; and Grönwall's inequality. The constant $C := 3K_{\tilde{b}}^2$ and $C(t) := tC \exp\left(\frac{1}{2}t^2 C\right)$. Thus, we have

$$\mathsf{d}^2(F_T(\Upsilon)_t, F_T(\Upsilon')_t) \leq C(t) \int_0^t \mathsf{d}^2(\Upsilon_s, \Upsilon_s')\,\mathrm{d}s.$$

Then, for $F_T^k$ denoting $k$ successive composition of $F_T$, one can inductively show that it satisfies

$$\mathsf{d}^2(F_T^k(\Upsilon)_t, F_T^k(\Upsilon')_t) \leq \frac{(tC(t))^k}{k!} \sup_{s \in [0,T]} \mathsf{d}^2(\Upsilon_s, \Upsilon_s').$$

Taking the supremum, we have

$$\sup_{s \in [0,T]} \mathsf{d}^2(F_T^k(\Upsilon)_s, F_T^k(\Upsilon')_s) \leq \frac{(TC(T))^k}{k!} \sup_{s \in [0,T]} \mathsf{d}^2(\Upsilon_s, \Upsilon_s').$$

Thus, for a large enough $k$, we have shown that $F_T^k$ is a contraction and from Banach Fixed Point Theorem and the completeness of the space $(C([0, T], \Theta \times \mathcal{P}(\mathbb{R}^{d_z})), \sup \mathsf{d})$, we have existence and uniqueness. $\qquad\square$

### F.3 Proof of Prop. 9

Recall, the process defined in Prop. 9:

$$\mathrm{d}\theta_t^{\gamma,M} = -\nabla_\theta \mathcal{E}_\lambda^\gamma(\theta_t^{\gamma,M}, r_t^{\gamma,M})\,\mathrm{d}t, \quad \text{where } r_t^{\gamma,M} = \frac{1}{M}\sum_{m=1}^M \delta_{Z_{t,m}^{\gamma,M}}$$

$$\forall m \in [M] : \mathrm{d}Z_{t,m}^{\gamma,M} = b^\gamma(\theta_t^{\gamma,M}, r_t^{\gamma,M}, Z_{t,m}^{\gamma,M})\,\mathrm{d}t + \sqrt{2\lambda_r}\,\mathrm{d}W_{t,m}.$$

and $\gamma$-PVI (defined in Eq. (17)) augmented with extra particles (in the sense that there are $M$ independent copies of the $Z$-process) to facilitate a synchronous coupling argument

$$\mathrm{d}\theta_t^\gamma = -\nabla_\theta \mathcal{E}_\lambda^\gamma(\theta_t^\gamma, \mathrm{Law}(Z_{t,1}^\gamma))\,\mathrm{d}t,$$

$$\forall m \in [M] : \mathrm{d}Z_{t,m}^\gamma = b^\gamma(\theta_t^\gamma, \mathrm{Law}(Z_{t,1}^\gamma), Z_{t,m}^\gamma)\,\mathrm{d}t + \sqrt{2\lambda_r}\,\mathrm{d}W_{t,m}.$$

*Proof of Prop. 9.* This is equivalent to proving that

$$\underbrace{\mathbb{E}\sup_{t\in[0,T]} \|\theta_t^\gamma - \theta_t^{\gamma,M}\|^2}_{(a)} + \underbrace{\mathbb{E}\sup_{t\in[0,T]} \left\{ \frac{1}{M}\sum_{m=1}^M \|Z_{t,m}^\gamma - Z_{t,m}^{\gamma,M}\|^2 \right\}}_{(b)} = o(1). \tag{24}$$

We shall treat the two terms individually. We begin with (a) in (24), where Jensen's inequality gives:

$$(a) \text{ in } (24) = \mathbb{E}\sup_{t\in[0,T]} \left\| \int_0^t \left[ \nabla_\theta \mathcal{E}_\lambda^\gamma(\theta_s^{\gamma,M}, r_s^{\gamma,M}) - \nabla_\theta \mathcal{E}_\lambda^\gamma(\theta_s^\gamma, r_s^\gamma) \right]\,\mathrm{d}s \right\|^2$$

$$\leq T\mathbb{E}\int_0^T \left\| \nabla_\theta \mathcal{E}_\lambda^\gamma(\theta_t^{\gamma,M}, r_t^{\gamma,M}) - \nabla_\theta \mathcal{E}_\lambda^\gamma(\theta_s^\gamma, r_s^\gamma) \right\|^2\,\mathrm{d}t$$

$$\leq C_\theta \int_0^T \mathbb{E}\|\theta_s^\gamma - \theta_s^{\gamma,M}\|^2 + \mathbb{E}\mathsf{W}_2^2(r_s^{\gamma,M}, r_s^\gamma)\,\mathrm{d}t. \tag{25}$$

where $C_\theta := 2TK_{\mathcal{E}_\lambda^\gamma}^2$, we apply Cauchy–Schwarz; and the $C_r$ inequality with the Lipschitz continuity of $\nabla_\theta \mathcal{E}_\lambda^\gamma$ from Prop. 12. Using the $C_r$ inequality again, together with the triangle inequality:

$$\mathbb{E}\mathsf{W}_2^2(r_s^{\gamma,M}, r_s^\gamma) \leq 2\mathbb{E}\mathsf{W}_2^2(r_s^\gamma, \hat{r}_s^\gamma) + 2\mathbb{E}\mathsf{W}_2^2(r_s^{\gamma,M}, \hat{r}_s^\gamma)$$

$$\leq o(1) + \frac{2}{M}\sum_{m=1}^M \mathbb{E}\|Z_{s,m}^\gamma - Z_{s,m}^{\gamma,M}\|^2, \tag{26}$$

where $\hat{r}_s^\gamma = \frac{1}{M}\sum_{m=1}^M \delta_{Z_{s,m}^\gamma}$ and we use Fournier and Guillin (2015). Note that we also have

$$\|\theta_s^\gamma - \theta_s^{\gamma,M}\|^2 \leq \sup_{s'\in[0,T]} \|\theta_{s'}^\gamma - \theta_{s'}^{\gamma,M}\|^2, \tag{27}$$

$$\frac{1}{M}\sum_{m=1}^M \|Z_{s,m}^\gamma - Z_{s,m}^{\gamma,M}\|^2 \leq \sup_{s'\in[0,T]} \frac{1}{M}\sum_{m=1}^M \|Z_{s',m}^\gamma - Z_{s',m}^{\gamma,M}\|^2. \tag{28}$$

Applying Eq. (26) in Eq. (25) then Eqs. (27) and (28), we obtain

$$(a) \leq 2C_\theta \int_0^T \mathbb{E}\sup_{s\in[0,T]} \|\theta_s^\gamma - \theta_s^{\gamma,M}\|^2 + \mathbb{E}\sup_{s\in[0,T]} \frac{1}{M}\sum_{m=1}^M \|Z_{s,m}^\gamma - Z_{s,m}^{\gamma,M}\|^2\,\mathrm{d}s + o(1). \tag{29}$$

Similarly, for (b) in (24), we have

$$(b) = \mathbb{E}\sup_{t\in[0,T]} \frac{1}{M}\sum_{m=1}^M \left\| \int_0^t b^\gamma(\theta_s^{\gamma,M}, r_s^{\gamma,M}, Z_{s,m}^{\gamma,M}) - b^\gamma(\theta_s^\gamma, \mathrm{Law}(Z_{s,1}^\gamma), Z_{s,m}^\gamma)\,\mathrm{d}s \right\|^2$$

$$\leq C_z \mathbb{E}\int_0^T \|\theta_s^{\gamma,M} - \theta_s^\gamma\|^2 + \mathsf{W}_2^2(r_s^{\gamma,M}, \mathrm{Law}(Z_{s,1}^\gamma)) + \frac{1}{M}\sum_{m=1}^M \|Z_{s,m}^\gamma - Z_{s,m}^{\gamma,M}\|^2\,\mathrm{d}s,$$

where $C_z := 3K_{b^\gamma}^2$ and, as before, we apply Cauchy–Schwarz, Lipschitz and $C_r$ inequalities. Then from Eqs. (26) to (28), we obtain

$$(b) \leq C\mathbb{E}\int_0^T \sup_{s\in[0,T]} \|\theta_s^{\gamma,M} - \theta_s^\gamma\|^2 + \sup_{s\in[0,T]} \frac{1}{M}\sum_{m=1}^M \|Z_{s,m}^\gamma - Z_{s,m}^{\gamma,M}\|^2 + o(1)\mathrm{d}s. \quad (30)$$

Combining Eqs. (29) and (30) and applying Grönwall's inequality, we obtain

$$\mathbb{E}\sup_{t\in[0,T]} \|\theta_t^\gamma - \theta_t^{\gamma,M}\|^2 + \mathbb{E}\sup_{t\in[0,T]} \left\{ \frac{1}{M}\sum_{m=1}^M \|Z_{t,m}^\gamma - Z_{t,m}^{\gamma,M}\|^2 \right\} = o(1).$$

Taking the limit, we have the desired result. $\qquad\square$

## F.4 The drift in Eq. (17) is Lipschitz

In this section, we show that the drift in the $\gamma$-PVI flow in Eq. (17) is Lipschitz.

**Proposition 11.** *Under the same assumptions as Prop. 8; the drift $\tilde{b}(A, r)$ is Lipschitz, i.e., there exists a constant $K_{\tilde{b}} \in \mathbb{R}_{>0}$ such that:*

$$\|\tilde{b}^\gamma(\theta, r, z) - \tilde{b}^\gamma(\theta', r', z')\| \leq K_{\tilde{b}}(\|(\theta, z) - (\theta', z')\| + \mathsf{W}_2(r, r')), \ \ \forall \theta, \theta' \in \Theta, z, z' \in \mathcal{Z}, r, r' \in \mathcal{P}(\mathcal{Z}).$$

*Proof.* From the definition and using the concavity of $\sqrt{\cdot}$ (which ensures that for any $a, b \geq 0$, $\sqrt{a+b} \leq \sqrt{a} + \sqrt{b}$), we obtain

$$\|\tilde{b}^\gamma(\theta, r, z) - \tilde{b}^\gamma(\theta', r', z')\| \leq \|\nabla_\theta \mathcal{E}_\lambda^\gamma(\theta, r) - \nabla_\theta \mathcal{E}_\lambda^\gamma(\theta', r')\| + \|b^\gamma(\theta, r, z) - b^\gamma(\theta', r', z')\|.$$

It is established below in Prop. 12 that $\nabla_\theta \mathcal{E}_\lambda^\gamma$ satisfies a Lipschitz inequality, i.e., there is some $K_{\mathcal{E}_\lambda^\gamma} \in \mathbb{R}_{>0}$ such that

$$\|\nabla_\theta \mathcal{E}_\lambda^\gamma(\theta, r) - \nabla_\theta \mathcal{E}_\lambda^\gamma(\theta', r')\| \leq K_{\mathcal{E}_\lambda^\gamma}(\|\theta - \theta'\| + \mathsf{W}_2(r, r')).$$

It is established below in Prop. 13 that $b^\gamma$ satisfies a Lipschitz inequality, i.e., there is some $K_{b^\gamma} \in \mathbb{R}_{>0}$ such that

$$\|b^\gamma(\theta, r, z) - b^\gamma(\theta', r', z')\| \leq K_{b^\gamma}(\|(\theta, z) - (\theta', z')\| + \mathsf{W}_2(r, r')).$$

Hence, we have obtained as desired with $K_{\tilde{b}} = K_{\mathcal{E}_\lambda^\gamma} + K_{b^\gamma}$. $\qquad\square$

**Proposition 12.** *Under the same assumptions as Prop. 8, the function $(\theta, r) \mapsto \nabla_\theta \mathcal{E}_\lambda^\gamma(\theta, r)$ is Lipschitz, i.e., there exist some constant $K_{\mathcal{E}_\lambda^\gamma} \in \mathbb{R}_{>0}$ such that*

$$\|\nabla_\theta \mathcal{E}_\lambda^\gamma(\theta, r) - \nabla_\theta \mathcal{E}_\lambda^\gamma(\theta', r')\| \leq K_{\mathcal{E}_\lambda^\gamma}(\|\theta - \theta'\| + \mathsf{W}_2(r, r')), \ \ \forall (\theta, r), (\theta', r') \in \mathcal{M}.$$

*Proof.* From the definition, we have

$$\nabla_\theta \mathcal{E}_\lambda^\gamma(\theta, r) = \nabla_\theta \mathcal{E}^\gamma(\theta, r) + \nabla_\theta \mathsf{R}_\lambda(\theta, r).$$

Thus, if both $\nabla_\theta \mathcal{E}^\gamma$ and $\nabla_\theta \mathsf{R}_\lambda$ are Lipschitz, then so is $\nabla_\theta \mathcal{E}_\lambda^\gamma$. Since $\mathsf{R}_\lambda$ has Lipschitz gradient (by Assumption 1), it remains to be shown that $\nabla_\theta \mathcal{E}^\gamma$ is Lipschitz. From Prop. 6, we have

$$\nabla_\theta \mathcal{E}^\gamma(\theta, r) = \mathbb{E}_{p_k(\epsilon)r(z)}\left[ (\nabla_\theta \phi_\theta \cdot [s_{\theta,r}^\gamma - s_p])(z, \epsilon) \right] = \int \nabla_\theta \phi_\theta \cdot d_{\theta,r}^{p;\gamma}(z, \epsilon)\, p_k(\mathrm{d}\epsilon)r(\mathrm{d}z),$$

where $d_{\theta,r}^{p;\gamma}(z, \epsilon) := s_{\theta,r}^\gamma(z, \epsilon) - s_p(z, \epsilon)$. Then, applying Jensen's inequality, we obtain

$$\|\nabla_\theta \mathcal{E}^\gamma(\theta, r) - \nabla_\theta \mathcal{E}^\gamma(\theta', r')\|$$
$$= \left\| \int p(\epsilon) \int \left[ \nabla_\theta \phi_\theta \cdot d_{\theta,r}^{p;\gamma}(z, \epsilon)r(z) - \nabla_\theta \phi_{\theta'} \cdot d_{\theta',r'}^{p;\gamma}(z, \epsilon)r'(z) \right] \mathrm{d}z\mathrm{d}\epsilon \right\|$$
$$\leq \int p(\epsilon) \left\| \int \left[ \nabla_\theta \phi_\theta \cdot d_{\theta,r}^{p;\gamma}(z, \epsilon)r(z) - \nabla_\theta \phi_{\theta'} \cdot d_{\theta',r'}^{p;\gamma}(z, \epsilon)r'(z) \right] \mathrm{d}z \right\| \mathrm{d}\epsilon. \quad (31)$$

Focusing on the integrand, we can upper-bound it with

$$\left\| \int \left[ \nabla_\theta \phi_\theta \cdot d_{\theta,r}^{p,\gamma}(z,\epsilon) r(z) - \nabla_\theta \phi_{\theta'} \cdot d_{\theta',r'}^{p,\gamma}(z,\epsilon) r'(z) \right] \mathrm{d}z \right\|$$

$$\overset{(a)}{\leq} \left\| \int \nabla_\theta \phi_\theta \cdot d_{\theta,r}^{p,\gamma}(z,\epsilon) [r(z) - r'(z)] \mathrm{d}z \right\| + \left\| \int \left[ \nabla_\theta \phi_\theta \cdot d_{\theta,r}^{p,\gamma}(z,\epsilon) - \nabla_\theta \phi_{\theta'} \cdot d_{\theta',r'}^{p,\gamma}(z,\epsilon) \right] r'(z) \mathrm{d}z \right\|$$

$$\overset{(b)}{\leq} \int \left\| \nabla_\theta \phi_\theta \cdot d_{\theta,r}^{p,\gamma}(z,\epsilon) \right\| |r(z) - r'(z)| \mathrm{d}z$$

$$+ \int \left\| \nabla_\theta \phi_\theta \cdot d_{\theta,r}^{p,\gamma}(z,\epsilon) - \nabla_\theta \phi_{\theta'} \cdot d_{\theta',r'}^{p,\gamma}(z,\epsilon) \right\| r'(z) \mathrm{d}z.$$

where in (a) we add and subtract the relevant terms and invoke the triangle inequality, and in (b) we apply Jensen's inequality. Plugging this back into Eq. (31), we obtain

$$\|\nabla_\theta \mathcal{E}^\gamma(\theta,r) - \nabla_\theta \mathcal{E}^\gamma(\theta',r')\|$$

$$\leq \int \mathbb{E}_{p_k(\epsilon)} \left\| \nabla_\theta \phi_\theta \cdot d_{\theta,r}^{p,\gamma}(z,\epsilon) \right\| |r(z) - r'(z)| \mathrm{d}z \tag{32}$$

$$+ \int \mathbb{E}_{p_k(\epsilon)} \left\| \nabla_\theta \phi_\theta \cdot d_{\theta,r}^{p,\gamma}(z,\epsilon) - \nabla_\theta \phi_{\theta'} \cdot d_{\theta',r'}^{p,\gamma}(z,\epsilon) \right\| r'(z) \mathrm{d}z, \tag{33}$$

where the interchange of integrals is justified from Fubini's theorem for non-negative functions (also known as Tonelli's theorem).

As we shall later show, the two terms have the following upper bounds:

$$(32) \leq K \mathsf{W}_1(r,r'), \text{ and} \tag{34}$$

$$(33) \leq K(\|\theta - \theta'\| + \mathsf{W}_1(r,r')), \tag{35}$$

where $K$ denotes a generic constant; and, upon noting that $\mathsf{W}_1 \leq \mathsf{W}_2$, we obtained the desired result. Now, we shall verify Eqs. (34) and (35). For the Eq. (34), we use the fact that the map $z \mapsto \mathbb{E}_{p_k(\epsilon)} \left\| \nabla_\theta \phi_\theta \cdot d_{\theta,r}^p(z,\epsilon) \right\|$ is Lipschitz then from the dual representation of $\mathsf{W}_1$, we obtain the desired result. To see that the aforementioned map is Lipschitz,

$$\left| \mathbb{E}_{p_k(\epsilon)} \left\| \nabla_\theta \phi_\theta \cdot d_{\theta,r}^{p,\gamma}(z,\epsilon) \right\| - \mathbb{E}_{p_k(\epsilon)} \left\| \nabla_\theta \phi_\theta \cdot d_{\theta,r}^{p,\gamma}(z',\epsilon) \right\| \right|$$

$$\overset{(a)}{\leq} \mathbb{E}_{p_k(\epsilon)} \left\| \nabla_\theta \phi_\theta \cdot d_{\theta,r}^{p,\gamma}(z,\epsilon) - \nabla_\theta \phi_\theta \cdot d_{\theta,r}^{p,\gamma}(z',\epsilon) \right\|$$

$$\overset{(b)}{\leq} \mathbb{E}_{p_k(\epsilon)} \left\| \nabla_\theta \phi_\theta(z,\epsilon) \cdot (d_{\theta,r}^{p,\gamma}(z,\epsilon) - d_{\theta,r}^{p,\gamma}(z',\epsilon)) \right\|$$

$$+ \mathbb{E}_{p_k(\epsilon)} \left\| (\nabla_\theta \phi_\theta(z,\epsilon) - \nabla_\theta \phi_\theta(z',\epsilon)) \cdot d_{\theta,r}^{p,\gamma}(z',\epsilon) \right\|$$

$$\overset{(c)}{\leq} \mathbb{E}_{p_k(\epsilon)} \left[ \|\nabla_\theta \phi_\theta(z,\epsilon)\|_F \left\| d_{\theta,r}^{p,\gamma}(z,\epsilon) - d_{\theta,r}^{p,\gamma}(z',\epsilon) \right\| \right]$$

$$+ \mathbb{E}_{p_k(\epsilon)} \left[ \|\nabla_\theta \phi_\theta(z,\epsilon) - \nabla_\theta \phi_\theta(z',\epsilon)\|_F \left\| d_{\theta,r}^{p,\gamma}(z',\epsilon) \right\| \right]$$

$$\overset{(d)}{\leq} \mathbb{E}_{p_k(\epsilon)} \left[ (a_\phi \|\epsilon\| + b_\phi)(a_d \|\epsilon\| + b_d) \right] \|z - z'\|$$

$$+ \mathbb{E}_{p_k(\epsilon)} \left[ (a_\phi \|\epsilon\| + b_\phi) \left\| d_{\theta,r}^{p,\gamma}(z',\epsilon) \right\| \right] \|z - z'\|,$$

$$\overset{(e)}{\leq} \mathbb{E}_{p_k(\epsilon)} \left[ (a_\phi \|\epsilon\| + b_\phi)(a_d \|\epsilon\| + b_d) \right] \|z - z'\|$$

$$+ \frac{1}{2} \mathbb{E}_{p_k(\epsilon)} \left[ (a_\phi \|\epsilon\| + b_\phi)^2 + \left\| d_{\theta,r}^{p,\gamma}(z',\epsilon) \right\|^2 \right] \|z - z'\|,$$

where (a) we use the reverse triangle inequality; (b) we add and subtract relevant terms and apply the triangle inequality; (c) we use a property of the matrix norm with $\| \cdot \|_F$ denoting the Frobenius norm; (d) we utilize Assumption 3 and the Lipschitz property from Prop. 15; (e) we apply Young's

inequality. Then, from the fact that

$$\mathbb{E}_{p_k(\epsilon)} \left[ (a_\phi \|\epsilon\| + b_\phi)(a_d \|\epsilon\| + b_d) \right] < \infty, \quad \mathbb{E}_{p_k(\epsilon)} \left[ (a_\phi \|\epsilon\| + b_\phi)^2 + \left\| d_{\theta,r}^{p,\gamma}(z', \epsilon) \right\|^2 \right] < \infty, \quad (36)$$

which holds from the assumption that $p_k$ has finite second moments Assumption 3, and from our assumption that $\mathbb{E}_{p_k(\epsilon)} \left\| d_{\theta,r}^{p,\gamma}(z', \epsilon) \right\|$ is bounded. Hence, the map is Lipschitz and so Eq. (34) holds.

As for Eq. (35), we focus on the integrand in Eq. (33)

$$\mathbb{E}_{p_k(\epsilon)} \left\| \nabla_\theta \phi_\theta \cdot d_{\theta,r}^{p,\gamma}(z, \epsilon) - \nabla_\theta \phi_{\theta'} \cdot d_{\theta',r'}^{p,\gamma}(z, \epsilon) \right\|$$

$$\leq \mathbb{E}_{p_k(\epsilon)} \left[ \left\| \nabla_\theta \phi_\theta \cdot (d_{\theta,r}^{p,\gamma} - d_{\theta',r'}^{p,\gamma})(z, \epsilon) \right\| + \left\| (\nabla_\theta \phi_\theta - \nabla_\theta \phi_{\theta'}) \cdot d_{\theta',r'}^p(z, \epsilon) \right\| \right]$$

$$\leq \mathbb{E}_{p_k(\epsilon)} \left[ \|\nabla_\theta \phi_\theta(z, \epsilon)\|_F \left\| (d_{\theta,r}^{p,\gamma} - d_{\theta',r'}^{p,\gamma})(z, \epsilon) \right\| + \|(\nabla_\theta \phi_\theta - \nabla_\theta \phi_{\theta'})(z, \epsilon)\|_F \left\| d_{\theta',r'}^{p,\gamma}(z, \epsilon) \right\| \right]$$

$$\leq \mathbb{E}_{p_k(\epsilon)} \left[ (a_\phi \|\epsilon\| + b_\phi)(a_d \|\epsilon\| + b_d) \right] \left( \|\theta - \theta'\| + W_1(r, r') \right)$$

$$+ \mathbb{E}_{p_k(\epsilon)} \left[ (a_\phi \|\epsilon\| + b_\phi) \left\| d_{\theta',r'}^{p,\gamma}(z, \epsilon) \right\| \right] \|\theta - \theta'\|,$$

where, for the last line, we apply Prop. 15 and Assumption 3. Applying Young's inequality and (36), we have the desired result. $\qquad \square$

**Proposition 13** ($b^\gamma$ is Lipschitz)**.** *Under the same assumptions as Prop. 8, the map $b^\gamma$ is $K_{b^\gamma}$-Lipschitz, i.e., there exists a constant $K_{b^\gamma} \in \mathbb{R}_{>0}$ such that the following inequality holds for all $(\theta, z, r), (\theta', z', r') \in \Theta \times \mathbb{R}^{d_z} \times \mathcal{P}(\mathbb{R}^{d_z})$:*

$$\|b^\gamma(\theta, r, z) - b^\gamma(\theta', r', z')\| \leq K_{b^\gamma} (\|(\theta, z) - (\theta', z')\| + W_1(r, r')).$$

*Proof.* One can write the drift $b^\gamma$ as follows (can be found in Eq. (43) similarly to Prop. 6), we have

$$b^\gamma(\theta, r, z) = -\mathbb{E}_{p_k(\epsilon)} \left[ (\nabla_z \phi_\theta \cdot [s_{\theta,r}^\gamma - s_p + \Gamma_{\theta,r}^\gamma])(z, \epsilon) \right] + \nabla_x \log p_0(z),$$

where $\Gamma_{\theta,r}^\gamma(z, \epsilon) := \frac{\gamma \nabla_x q_{\theta,r}(\phi_\theta(z, \epsilon))}{(q_{\theta,r}(\phi_\theta(z, \epsilon)) + \gamma)^2}$. Hence,

$$\|b^\gamma(\theta, r, z) - b^\gamma(\theta', r', z')\| \leq \|\mathbb{E}_{p_k(\epsilon)}[(\nabla_z \phi_\theta \cdot [d_{\theta,r}^{p,\gamma} + \Gamma_{\theta,r}^\gamma])(z, \epsilon) - (\nabla_z \phi_{\theta'} \cdot [d_{\theta',r'}^p + \Gamma_{\theta',r'}^\gamma])(z', \epsilon)]\|$$

$$+ \|\nabla_z \log p_0(z) - \nabla_z \log p_0(z')\|$$

$$\leq \mathbb{E}_{p_k(\epsilon)} \|(\nabla_z \phi_\theta \cdot d_{\theta,r}^{p,\gamma} + \Gamma_{\theta,r}^\gamma)(z, \epsilon) - (\nabla_z \phi_{\theta'} \cdot [d_{\theta',r'}^{p,\gamma} + \Gamma_{\theta',r'}^\gamma])(z', \epsilon)\|$$

$$+ K_{p_0} \|z - z'\|,$$

where for the last inequality we use Jensen's inequality and Assumption 1. Since we have

$$\mathbb{E}_{p_k(\epsilon)} \|(\nabla_z \phi_\theta \cdot [d_{\theta,r}^{p,\gamma} + \Gamma_{\theta,r}^\gamma)](z, \epsilon) - (\nabla_z \phi_{\theta'} \cdot [d_{\theta',r'}^{p,\gamma} + \Gamma_{\theta',r'}^\gamma])(z', \epsilon)\|$$

$$\overset{(a)}{\leq} \mathbb{E}_{p_k(\epsilon)} \|(\nabla_z \phi_\theta \cdot [d_{\theta,r}^{p,\gamma} + \Gamma_{\theta,r}^\gamma])(z, \epsilon) - \nabla_z \phi_\theta(z, \epsilon) \cdot [d_{\theta',r'}^{p,\gamma} + \Gamma_{\theta',r'}^\gamma](z', \epsilon)\|$$

$$+ \mathbb{E}_{p_k(\epsilon)} \|\nabla_z \phi_\theta(z, \epsilon) \cdot [d_{\theta',r'}^{p,\gamma} + \Gamma_{\theta',r'}^\gamma](z', \epsilon) - (\nabla_z \phi_{\theta'} \cdot [d_{\theta',r'}^{p,\gamma} + \Gamma_{\theta',r'}^\gamma])(z', \epsilon)\|$$

$$\overset{(b)}{\leq} \mathbb{E}_{p_k(\epsilon)} \|\nabla_z \phi_\theta(z, \epsilon)\|_F \|(d_{\theta,r}^{p,\gamma} + \Gamma_{\theta,r}^\gamma)(z, \epsilon) - (d_{\theta',r'}^{p,\gamma} + \Gamma_{\theta',r'}^\gamma)(z', \epsilon)\|$$

$$+ \mathbb{E}_{p_k(\epsilon)} \|\nabla_z \phi_\theta(z, \epsilon) - \nabla_z \phi_{\theta'}(z', \epsilon)\|_F \|(d_{\theta',r'}^{p,\gamma} + \Gamma_{\theta',r'}^\gamma)(z', \epsilon)\|$$

$$\overset{(c)}{\leq} \mathbb{E}_{p_k(\epsilon)} (a_\phi \|\epsilon\| + b_\phi)(\|d_{\theta,r}^{p,\gamma}(z, \epsilon) - d_{\theta',r'}^{p,\gamma}(z', \epsilon)\| + \|\Gamma_{\theta,r}^\gamma(z, \epsilon) - \Gamma_{\theta',r'}^\gamma(z', \epsilon)\|) \quad (37)$$

$$+ \mathbb{E}_{p_k(\epsilon)} (a_\phi \|\epsilon\| + b_\phi) \|(d_{\theta',r'}^{p,\gamma} + \Gamma_{\theta',r'}^\gamma)(z', \epsilon)\| \|(\theta, z) - (\theta', z')\| \quad (38)$$

where, for (a), we add and subtract the relevant terms and invoke the triangle inequality, in (b) we use properties of the matrix norm, and in (c) we use the bounded gradient and Lipschitz gradient in Assumption 3. For Eq. (37); upon using Props. 14 and 15, which are established below, we obtain

$$\mathbb{E}_{p_k(\epsilon)} (a_\phi \|\epsilon\| + b_\phi)(\|d_{\theta,r}^{p,\gamma}(z, \epsilon) - d_{\theta',r'}^{p,\gamma}(z', \epsilon)\| + \|\Gamma_{\theta,r}^\gamma(z, \epsilon) - \Gamma_{\theta',r'}^\gamma(z', \epsilon)\|)$$

$$\leq \mathbb{E}_{p_k(\epsilon)} (a_\phi \|\epsilon\| + b_\phi)[(a_d + a_\Gamma) \|\epsilon\| + (b_d + b_\Gamma)](\|(\theta, z) - (\theta', z')\| + W_1(r, r')). \quad (39)$$

As for the second term, Eq. (38),

$$\mathbb{E}_{p_k(\epsilon)}(a_\phi\|\epsilon\| + b_\phi)\|(d^{p;\gamma}_{\theta',r'} + \Gamma^\gamma_{\theta',r'})(z',\epsilon)\|$$

$$\overset{(a)}{\leq} \mathbb{E}_{p_k(\epsilon)}(a_\phi\|\epsilon\| + b_\phi)[\|d^{p;\gamma}_{\theta',r'}(z',\epsilon)\| + \|\Gamma^\gamma_{\theta',r'}(z',\epsilon)\|]$$

$$\overset{(b)}{\leq} \mathbb{E}_{p_k(\epsilon)}(a_\phi\|\epsilon\| + b_\phi)[\|d^{p;\gamma}_{\theta',r'}(z',\epsilon)\| + B_\Gamma]$$

$$\overset{(c)}{\leq} \mathbb{E}_{p_k(\epsilon)}\frac{1}{2}(a_\phi\|\epsilon\| + b_\phi)^2 + \frac{1}{2}\|d^{p;\gamma}_{\theta',r'}(z',\epsilon)\|^2 + B_\Gamma(a_\phi\|\epsilon\| + b_\phi) \tag{40}$$

where (a) follows from the triangle inequality, (b) we use Prop. 14 boundedness of $\Gamma$, (c) we apply Young's inequality to the first term. Similarly to Eq. (36), from our Assumption 3 and our boundedness assumption of the score, we have as desired. Combining Eq. (39) with the result of plugging Eq. (40) into Eq. (38), we obtain the result. $\qquad\square$

**Proposition 14** ($\Gamma$ is Lipschitz and bounded)**.** *Under Assumption 2, the map $(\theta, r, z) \mapsto \Gamma^\gamma_{\theta,r}(z,\epsilon)$ is Lipschitz and bounded. (Lipschitz) there are constants $a_\Gamma, b_\Gamma \in \mathbb{R}_{>0}$ such that following hold:*

$$\|\Gamma^\gamma_{\theta,r}(z,\epsilon) - \Gamma^\gamma_{\theta,r}(z,\epsilon)\| \leq (a_\Gamma\|\epsilon\| + b_\Gamma)(\|(\theta,z) - (\theta',z')\| + \mathsf{W}_1(r,r')).$$

*Furthermore, it is bounded*

$$\|\Gamma^\gamma_{\theta,r}(z,\epsilon)\| \leq B_\Gamma.$$

*Proof.* Since $\Gamma^\gamma_{\theta,r} = \frac{\gamma\nabla_x \log(q_{\theta,r}(x)+\gamma)}{q_{\theta,r}(x)+\gamma}$, where $x := \phi_\theta(z,\epsilon)$, and $x' := \phi_{\theta'}(z',\epsilon)$, we have:

$$\|\Gamma^\gamma_{\theta,r}(z,\epsilon) - \Gamma^\gamma_{\theta',r'}(z',\epsilon)\|$$

$$\leq \gamma\left\|\frac{(q_{\theta',r'}(x')+\gamma)\nabla_x\log(q_{\theta,r}(x)+\gamma) - (q_{\theta,r}(x)+\gamma)\nabla_x\log(q_{\theta',r'}(x')+\gamma)}{(q_{\theta,r}(x)+\gamma)(q_{\theta',r'}(x')+\gamma)}\right\|$$

$$\leq \frac{1}{\gamma}|q_{\theta',r'}(x') - q_{\theta,r}(x)|\|\nabla_x\log(q_{\theta,r}(x)+\gamma)\|$$

$$+ \frac{1}{\gamma}(q_{\theta,r}(x)+\gamma)\|\nabla_x\log(q_{\theta,r}(x)+\gamma) - \nabla_x\log(q_{\theta',r'}(x')+\gamma)\|$$

$$\leq \frac{B_k}{\gamma^2}|q_{\theta',r'}(x') - q_{\theta,r}(x)| + \frac{(B_k+\gamma)}{\gamma}\|s^\gamma_{\theta,r}(z,\epsilon) - s^\gamma_{\theta',r'}(z',\epsilon)\|$$

$$\leq \frac{B_kK_q}{\gamma^2}(1 + a_\phi\|\epsilon\| + b_\phi)(\|(\theta,z) - (\theta',z')\| + \mathsf{W}_1(r,r'))$$

$$+ \frac{B_k+\gamma}{\gamma^2}(a_s\|\epsilon\| + b_s)(\|(\theta,z) - (\theta',z')\| + \mathsf{W}_1(r,r')),$$

where the last inequality follows from applying Prop. 18 and Assumption 3 to the first term and Prop. 16 to the last term . Hence, we have as desired.

Boundedness follows from the fact that $\left\|\frac{\gamma\nabla_xq_{\theta,r}(\phi_\theta(z,\epsilon))}{(q_{\theta,r}(\phi_\theta(z,\epsilon))+\gamma)^2}\right\| \leq \frac{1}{\gamma}\|\nabla_xq_{\theta,r}(\phi_\theta(z,\epsilon))\| \leq \frac{B_k}{\gamma}.$ $\qquad\square$

**Proposition 15.** *Under Assumptions 1 to 3, the map $(\theta, r, z) \mapsto s^\gamma_{\theta,r}(z,\epsilon) - s_p(z,\epsilon) =: d^{p;\gamma}_{\theta,r}(z,\epsilon)$ satisfies the following: there exist $a_d, b_d \in \mathbb{R}_{>0}$ such that for all $(\theta,r), (\theta',r') \in \mathcal{M}$, and $z, z' \in \mathbb{R}^{d_z}$, we have*

$$\|d^{p;\gamma}_{\theta,r}(z,\epsilon) - d^{p;\gamma}_{\theta',r'}(z',\epsilon)\| \leq (a_d\|\epsilon\| + b_d)(\|(\theta,z) - (\theta',z')\| + \mathsf{W}_1(r,r')).$$

*Proof.* Let $x := \phi_\theta(z,\epsilon)$, and $x' := \phi_{\theta'}(z',\epsilon)$. Then, we have

$$\|d^{p;\gamma}_{\theta,r}(z,\epsilon) - d^{p;\gamma}_{\theta',r'}(z',\epsilon)\| \leq \|\nabla_x\log p(x,y) - \nabla_x\log p(x',y)\|$$

$$+ \|\nabla_x\log(q_{\theta,r}(x)+\gamma) - \nabla_x\log(q_{\theta',r'}(x')+\gamma)\|$$

$$\leq K_p\|x - x'\| + \|\nabla_x\log(q_{\theta,r}(x)+\gamma) - \nabla_x\log(q_{\theta',r'}(x')+\gamma)\|$$

$$\leq K_p(a_\phi\|\epsilon\| + b_\phi)\|(\theta,z) - (\theta',z')\|$$

$$+ (a_s\|\epsilon\| + b_s)(\|(\theta,z) - (\theta',z')\| + \mathsf{W}_1(r,r')),$$

where Prop. 16 and Assumptions 1 and 3 are used. $\qquad\square$

**Proposition 16** ($s$ is Lipschitz)**.** *Under Assumptions 2 and 3 and $\gamma > 0$, the map $(\theta, r, z) \mapsto s^\gamma_{\theta,r}(z, \epsilon)$ satisfies the following: there exist constants $a_s, b_s \in \mathbb{R}_{>0}$ such that the following inequality holds for all $(\theta, r), (\theta', r') \in \mathcal{M}$, and $z, z' \in \mathbb{R}^{d_z}$:*

$$\|s^\gamma_{\theta,r}(z, \epsilon) - s^\gamma_{\theta',r'}(z', \epsilon)\| \leq (a_s\|\epsilon\| + b_s)(\|(\theta, z) - (\theta', z')\| + \mathsf{W}_1(r, r')),$$

*Proof.* For brevity, we write $x = \phi_\theta(z, \epsilon)$ and $x' = \phi_{\theta'}(z', \epsilon)$; from the definition, we have

$$
\begin{aligned}
\|s^\gamma_{\theta,r}(z, \epsilon) - s^\gamma_{\theta',r'}(z', \epsilon)\| &= \left\| \frac{\nabla_x q_{\theta,r}(x)}{q_{\theta,r}(x) + \gamma} - \frac{\nabla_x q_{\theta',r'}(x')}{q_{\theta',r'}(x') + \gamma} \right\| \\
&\overset{(a)}{\leq} \left\| \frac{\nabla_x q_{\theta,r}(x)}{q_{\theta,r}(x) + \gamma} - \frac{\nabla_x q_{\theta',r'}(x')}{q_{\theta,r}(x) + \gamma} \right\| + \left\| \frac{\nabla_x q_{\theta',r'}(x')}{q_{\theta,r}(x) + \gamma} - \frac{\nabla_x q_{\theta',r'}(x')}{q_{\theta',r'}(x') + \gamma} \right\| \\
&\leq \frac{1}{q_{\theta,r}(x) + \gamma} \|\nabla_x q_{\theta,r}(x) - \nabla_x q_{\theta',r'}(x')\| \\
&\quad + \|\nabla_x q_{\theta',r'}(x')\| \left| \frac{1}{q_{\theta,r}(x) + \gamma} - \frac{1}{q_{\theta',r'}(x') + \gamma} \right| \\
&\overset{(b)}{\leq} \frac{1}{\gamma} \|\nabla_x q_{\theta,r}(x) - \nabla_x q_{\theta',r'}(x')\| + \frac{B_k}{\gamma^2} |q_{\theta,r}(x) - q_{\theta',r'}(x')|, \qquad (41)
\end{aligned}
$$

where (a) we add and subtract the relevant terms and the triangle inequality; (b) we use the fact that $\|\nabla_x q_{\theta',r'}(x')\| = \|\int \nabla_x k_{\theta'}(x'|z)r'(\mathrm{d}z)\| \leq B_k$ (from Cauchy–Schwartz and the boundedness part of Assumption 2). Now, we deal with the terms individually. For the first term in Eq. (41), we use the fact that the map $(\theta, r, z) \mapsto \nabla_x q_{\theta,r}(\phi_\theta(z, \epsilon))$ is $K_q$-Lipschitz from Prop. 17. As for the second term in Eq. (41), we apply Prop. 18.

Hence, we obtain

$$
\begin{aligned}
\|s^\gamma_{\theta,r}(z, \epsilon) - s^\gamma_{\theta',r'}(z', \epsilon)\| &\leq \left( \frac{K_{gq}}{\gamma} + \frac{B_k K_q}{\gamma^2} \right) (\|(\theta, x) - (\theta', x')\| + \mathsf{W}_1(r, r')) \\
&\leq \left( \frac{K_{gq}}{\gamma} + \frac{B_k K_q}{\gamma^2} \right) (1 + a_\phi\|\epsilon\| + b_\phi)(\|(\theta, z) - (\theta', z')\| + \mathsf{W}_1(r, r')),
\end{aligned}
$$

where we use Assumption 3 for the last line. $\qquad \square$

**Proposition 17.** *Under Assumption 2, the map $(\theta, r, x) \mapsto \nabla_x q_{\theta,r}(x)$ is Lipschitz, i.e., for all $\epsilon$, there exists a $K_{gq} \in \mathbb{R}_{>0}$ such that the following inequality holds for all $(\theta, r), (\theta', r') \in \mathcal{M}$ and $z, z' \in \mathbb{R}^{d_z}$,*

$$\|\nabla_x q_{\theta,r}(x) - \nabla_x q_{\theta',r'}(x')\| \leq K_{gq}(\|(\theta, x) - (\theta', x')\| + \mathsf{W}_1(r, r')).$$

*Proof.* From direct computation,

$$
\begin{aligned}
\|\nabla_x q_{\theta,r}(x) - \nabla_x q_{\theta',r'}(x')\| &= \left\| \int [\nabla_x k_\theta(x|z)r(z) - \nabla_x k_{\theta'}(x'|z)r'(z)]\, \mathrm{d}z \right\| \\
&\overset{(a)}{\leq} \int \|\nabla_x [k_\theta(x|z) - k_{\theta'}(x'|z)]\|\, r(z)\, \mathrm{d}z \\
&\quad + \int \|\nabla_x k_{\theta'}(x'|z)\|\, |r - r'|(z)\, \mathrm{d}z \\
&\overset{(b)}{\leq} K_{gq}(\|(\theta, x) - (\theta', x')\| + \mathsf{W}_1(r, r')),
\end{aligned}
$$

where (a) we add and subtract the appropriate terms, apply the triangle inequality and the Cauchy-Schwarz inequality; (b) for the first term, we use the Lipschitz gradient Assumption 2; and for the second term, we use the dual representation of $\mathsf{W}_1$ with the fact map $z \mapsto \|\nabla_x \log k_\theta(x|z)\|$ is $K_k$-Lipschitz from the reverse triangle inequality and the Lipschitz Assumption 2. $\qquad \square$

**Proposition 18.** *Under Assumption 2, the map $(\theta, r, x) \mapsto q_{\theta,r}(x)$ is Lipschitz, i.e., there exists some $K_q \in \mathbb{R}_{>0}$ such that for all $(\theta, r, x), (\theta', r', x') \in \Theta \times \mathcal{P}(\mathbb{R}^{d_z}) \times \mathbb{R}^{d_x}$, we have*

$$|q_{\theta,r}(x) - q_{\theta',r'}(x')| < K_q(\|(\theta, x) - (\theta', x')\| + \mathsf{W}_1(r, r')).$$

*Proof.* From direct computation, we have

$$
\begin{aligned}
|q_{\theta,r}(x) - q_{\theta',r'}(x')| &\leq |q_{\theta,r}(x) - q_{\theta,r'}(x')| + |q_{\theta,r'}(x) - q_{\theta',r'}(x')| \\
&\leq \int |k_\theta(x|z)||r - r'|(z)\mathrm{d}z + \int |k_\theta(x|z) - k_{\theta'}(x'|z)|r(z)\mathrm{d}z \\
&\overset{(a)}{\leq} K_q(\mathsf{W}_1(r, r') + \|(\theta, x) - (\theta, x')\|)
\end{aligned}
$$

(42)

where (a) for the first term, we use the fact that the map $z \mapsto |k_\theta(x|z)|$ is $B_k$-Lipschitz (from the bounded gradient of Assumption 2), and again the Lipschitz property of $k$ from the same assumption. $\qquad\square$

# G  Algorithmic details

## G.1  Gradient estimator

*Proof of Prop. 6.* We show the derivation of the estimators in Eq. (10). Eq. (9) will follow similarly. We have

$$
\begin{aligned}
\nabla_z \delta_r \mathcal{E}[\theta, r](z) &= \nabla_z \mathbb{E}_{k_\theta(x|z)}\left[\log \frac{q_{\theta,r}(x)}{p(y, x)}\right] \\
&= \nabla_z \mathbb{E}_{\epsilon \sim p_k}\left[\log \frac{q_{\theta,r}(\phi_\theta(z, \epsilon))}{p(y, \phi_\theta(z, \epsilon))}\right].
\end{aligned}
$$

Assuming that $\phi_\theta$ and $p_k$ are sufficiently regular to justify the interchange of differentiation and integration, we obtain

$$
\nabla_z \delta_r \mathcal{E}[\theta, r](z) = \mathbb{E}_{\epsilon \sim p_k}\left[\nabla_z \log \frac{q_{\theta,r}(\phi(z, \epsilon))}{p(y, \phi(z, \epsilon))}\right].
$$

To obtain as desired, one can apply the chain rule. $\qquad\square$

Similarly, one can derive a Monte Carlo gradient estimator for $\nabla_z \delta_r \mathcal{E}^\gamma$ as follows:

$$
\begin{aligned}
\nabla_z \delta_r \mathcal{E}^\gamma[\theta, r](z) &= \nabla_z \mathbb{E}_{k_\theta(x|z)}\left[\log \frac{q_{\theta,r}(x) + \gamma}{p(y, x)} + \frac{q_{\theta,r}(x)}{q_{\theta,r}(x) + \gamma}\right] \\
&= \nabla_z \mathbb{E}_{\epsilon \sim p_k}\left[\log \frac{q_{\theta,r}(\phi_\theta(z, \epsilon)) + \gamma}{p(y, \phi_\theta(z, \epsilon))} + \frac{q_{\theta,r}(\phi_\theta(z, \epsilon))}{q_{\theta,r}(\phi_\theta(z, \epsilon)) + \gamma}\right].
\end{aligned}
$$

As before, if $\phi_\theta$ and $p_k$ are sufficiently regular, we obtain

$$
\nabla_z \delta_r \mathcal{E}^\gamma[\theta, r](z) = \mathbb{E}_{\epsilon \sim p_k}\left[\nabla_z \log \frac{q_{\theta,r}(\phi(z, \epsilon)) + \gamma}{p(y, \phi(z, \epsilon))} + \frac{\gamma \nabla q_{\theta,r}(\phi_\theta(z, \epsilon))}{(q_{\theta,r}(\phi_\theta(z, \epsilon)) + \gamma)^2}\right].
$$

(43)

To obtain as desired, one can apply chain rule.

## G.2  Preconditioners

Recall that the preconditioned gradient flow is given by

$$
\mathrm{d}\theta_t = -\Psi_t^\theta \nabla_\theta \mathcal{E}_\lambda(\theta_t, r_t)\,\mathrm{d}t, \quad \partial_t r_t = \nabla_z \cdot (r_t \Psi_t^r \nabla_z \delta_r \mathcal{E}_\lambda[\theta_t, r_t]),
$$

where $\delta_r \mathcal{E}_\lambda[\theta, r] = \delta_r \mathcal{E}[\theta, r] + \log r/p_0$ We can rewrite the dynamics of $r_t$ as

$$
\begin{aligned}
\partial_t r_t &= \nabla_z \cdot (r_t \Psi_t^r \nabla_z[\delta_r \mathcal{E}[\theta_t, r_t] - \log p_0 + \log r_t]), \\
&= \nabla_z \cdot (r_t \Psi_t^r \nabla_z[\delta_r \mathcal{E}[\theta_t, r_t] - \log p_0]) + \nabla_z \cdot (r_t \Psi_t^r \nabla_z \log r_t).
\end{aligned}
$$

The second term can be written as

$$
\nabla_z \cdot (r_t \Psi_t^r \nabla_z \log r_t) = \nabla_z \cdot (\Psi_t^r \nabla_z r_t) = \nabla_z \cdot (\nabla_z \cdot [\Psi_t^r r_t]) - \nabla_z \cdot (r_t \nabla_z \cdot \Psi_t^r)
$$

| Layers | Size |
|---|---|
| Input | $d_{in}$ |
| Linear($d_{in}, d_h$), LReLU | $d_h$ |
| Linear($d_h, d_h$), LReLU | $d_h$ |
| Linear($d_h, d_{out}$), | $d_{out}$ |

Table 3: Neural network architecture defined by $\mathrm{NN}(d_{in}, d_h, d_{out})$.

where $(\nabla \cdot \Psi_t^r)_i = \sum_{j=1}^{d_z} \partial_{z_j}[(\Psi_t^r)_{ij}]$, and last equality holds since

$$\sum_{i=1}^{d_z} \partial_{z_i} \left\{ \sum_{j=1}^{d_z} (\Psi_t^r)_{ij} \partial_{z_j} r_t \right\} = \sum_{i=1}^{d_z} \partial_{z_i} \left\{ \sum_{j=1}^{d_z} \left( \partial_{z_j}[(\Psi_t^r)_{ij} r_t] - r_t \partial_{z_j}[(\Psi_t^r)_{ij}] \right) \right\}.$$

Hence, we have the following dynamics of $r_t$:

$$\partial_t r_t = \nabla_z \cdot \left( r_t \left( \Psi_t^r \nabla_z [\delta_r \mathcal{E}[\theta_t, r_t] - \log p_0] - \nabla_z \cdot \Psi_t^r \right) \right) + \nabla_z \cdot \left( \nabla_z \cdot [\Psi_t^r r_t] \right)$$

**Examples.** Following in the essence of RMSProp (Tieleman and Hinton, 2012), we utilize the preconditioner defined as follows:

$$B_k = \beta B_{k-1} + (1-\beta)\mathrm{Diag}(A(\{\nabla_z \delta_r \mathcal{E}[\theta_k, r_k](Z_m)^2\}_{m=1}^M))$$
$$\Psi_k^r(Z) = (B_k)^{-0.5}$$

where $B_k \in \mathbb{R}^{d_z \times d_z}$ and $A$ is some aggregation function such as the mean or max. The idea is to normalize by the aggregated gradient of the first variation across all the particles since this is the dominant component in the drift of PVI. Similarly to RMSProp, it keeps an exponential moving average of the squared gradient which can then be used in the preconditioner.

# H Experimental details

In this section, we highlight additional details for reproducibility and computation. The code was written in JAX (Bradbury et al., 2018) and executed on a NVIDIA GeForce RTX 4090.

## H.1 Section 5.1

**Hyperparameters**. For the neural network, we use $f_\theta = \mathrm{NN}(2, 128, 2)$ defined in Table 3, the number of particles $M = 100$, $d_z = 2$, $K = 1000$, $h_\theta = 10^{-4}$, $h_z = 10^{-2}$, $\lambda_r = 10^{-8}$, for $\Psi^\theta$ we use RMSProp and we set $\Psi^r = I_2$.

**Computation Time**. Each run took 8 seconds using JIT compilation.

## H.2 Section 5.2

In this section, we outline all the experimental details regarding Section 5.2.

**Densities.** Table 4 shows the densities used in the toy experiments.

| Name | Density |
|---|---|
| Banana | $\mathcal{N}(x_2; x_1^2/4, 1)\mathcal{N}(x_1; 0, 2)$ |
| X-Shape | $\frac{1}{2}\mathcal{N}\left(0, \begin{pmatrix} 2 & 1.8 \\ 1.8 & 2 \end{pmatrix}\right) + \frac{1}{2}\mathcal{N}\left(0, \begin{pmatrix} 2 & -1.8 \\ -1.8 & 2 \end{pmatrix}\right)$ |
| Multimodal | $\frac{1}{8}\mathcal{N}\left(\begin{pmatrix} 2 \\ 2 \end{pmatrix}, I\right) + \frac{1}{8}\mathcal{N}\left(\begin{pmatrix} -2 \\ -2 \end{pmatrix}, I\right) + \frac{1}{2}\mathcal{N}\left(\begin{pmatrix} 2 \\ -2 \end{pmatrix}, I\right) + \frac{1}{4}\mathcal{N}\left(\begin{pmatrix} -2 \\ 2 \end{pmatrix}, I\right)$ |

Table 4: Densities used in toy experiments (see Section 5.2).

**Hyperparameters.** We set the number of parameter updates and particle steps to be $K = 15000$, and $d_z = 2$.

- $f_\theta$. We use $f_\theta = \mathrm{NN}(2, 512, 2)$.

- **PVI**. We use $M = 100$, $\lambda_\theta = 0$, $\lambda_r = 10^{-8}$, $h_x = 10^{-2}$, $h_\theta = 10^{-4}$, $\Psi^\theta$ we use the RMSProp preconditoner, $\Psi^r = I_{d_z}$, and $L = 250$.

- **SVI**. We use $K = 50$ to estimate the objective (Yin and Zhou, 2018, see Eq. (5)) which are around the values used in Yin and Zhou (2018). We utilize RMSProp with step size $10^{-4}$, and $r = \mathcal{N}(0, I_{d_z})$. The implicit distribution is set to $r = \mathcal{N}(0, I_{d_z})$.

- **UVI**. For the HMC sampler, we follow in (Titsias and Ruiz, 2019) and use 50 burn-in steps, with step-size $10^{-1}$ and 5 leap-frog steps. We use the RMSProp optimizer with stepsize $10^{-4}$ for $k_\theta$. The implicit distribution is set to $r = \mathcal{N}(0, I_{d_z})$.

- **SM**. For the "dual" function written as $f$ in the original paper (Yu and Zhang, 2023, see Algorithm 1) we use $\mathrm{NN}(2, 512, 2)$. We utilize RMSProp with decaying learning rate from $10^{-4}$ to $10^{-5}$ to optimize the kernel $k_\theta$, and RMSProp with $10^{-3}$ to $10^{-4}$ for the dual function $f$. The implicit distribution is set to $r = \mathcal{N}(0, I_{d_z})$.

**Sliced Wasserstein Distance.** We report the average sliced Wasserstein distance using 100 projections computed from 10000 samples from the target and the variational distribution.

**Two-Sample Test.** We use the MMD-Fuse implementation found in `https://github.com/antoninschrab/mmdfuse.git`.

**Computation Time.** An example run on Banana with JIT compilation, PVI took 42 seconds, UVI took 10 minutes 36 seconds, SM took 45 seconds, and SVI took 38 seconds.

## H.3   Section 5.3

In this section, we outline all the hyperparameters for each method used.

**Hyperparameters.** We use $K = 20000$ set $d_z = 10$. For all kernel parameters, we use RMSProp preconditioner with step size $h_\theta = 10^{-3}$ .

- $f_\theta$. We use $f_\theta = \mathrm{NN}(d_z, 512, 22)$.

- **PVI**. We use $M = 100$, $\lambda_\theta = 0$, $\lambda_r = 10^{-8}$, $h_x = 10^{-2}$, and for $\Psi^r$ we use the one described in App. G.2 with mean as the aggregate function.

- **SVI**. We use $K = 50$ to estimate the objective (Yin and Zhou, 2018, see Eq. (5)) which are around the values used in Yin and Zhou (2018).

- **UVI**. For the HMC sampler, we follow in (Titsias and Ruiz, 2019) and use 50 burn-in steps, with step-size $10^{-1}$ and 5 leap-frog steps.

- **SM**. We were unable to improve the performance of SM with our chosen kernel and instead used the implementation in `https://github.com/longinYu/SIVISM?utm_source=catalyzex.com` to obtain posterior samples with implementation details found in the code repository and in the paper (Yu and Zhang, 2023).

## H.4   Section 5.4

In this section, we outline all the experimental details regarding Section 5.4.

**Model.** We consider the neural network $\mathrm{BNN}(d_{in}^{\mathrm{bnn}}, d_h^{\mathrm{bnn}})$ defined as $f_x(o) = W_2^\top \mathrm{ReLU}(W_1^\top o + b_1) + b_2$ where $o \in \mathbb{R}^{d_{in}^{\mathrm{bnn}}}$, $x = [\mathrm{vec}(W_2), b_2, \mathrm{vec}(W_1), b_1]^\top$, $W_2 \in \mathbb{R}^{d_h^{\mathrm{bnn}} \times 1}$, $b_2 \in \mathbb{R}$, $W_1 \in \mathbb{R}^{d_{in}^{\mathrm{bnn}} \times d_h^{\mathrm{bnn}}}$, $b_1 \in \mathbb{R}^{d_h^{\mathrm{bnn}}}$. Given an input-output pair $\boldsymbol{Y} := \{(O_i, Y_i)\}_{i=1}^B$, the model can be defined as $p(\boldsymbol{Y}, x) = p(\boldsymbol{Y}|x)p(x)$ where the likelihood is $p(\boldsymbol{Y}|x) = \prod_{i=1}^B \mathcal{N}(Y_i; f_x(O_i), 0.01^2)$ and the prior is $\mathcal{N}(x; 0, 25I)$.

**Datasets.** For all the datasets, we standardize by removing the mean and dividing by the standard deviation.

- **Protein**. For the model, we use $\mathrm{BNN}(9, 30)$ which results in the problem having dimension $d_x = 331$. The dataset is composed of 1600 train examples, 401 test examples.

- **Yacht**. For the model, we use $\text{BNN}(6, 10)$ which results in the problem having dimension $d_x = 81$. The dataset is composed of $246$ train examples and $62$ test examples.
- **Concrete** For the model, we use $\text{BNN}(8, 10)$ which results in the problem having dimension $d_x = 101$. The dataset comprises of $824$ training examples and $206$ test examples.

**Hyperparameters.** We use $K = 1500$ set $d_z = 10$. For all kernel parameters, we use RMSProp preconditioner with step size $h_\theta = 10^{-3}$ that decays to $10^{-5}$ following a constant schedule that transitions every $100$ parameters steps.

- $f_\theta, \sigma_\theta$. We use $f_\theta = \text{NN}(d_z, 512, d_x)$ and $\sigma_\theta = \text{Softplus}(\text{NN}(d_z, 512, d_x)) + 10^{-8}$ and they share parameters except for the last layers.
- **PVI**. We use $M = 100$, $\lambda_\theta = 0$, $\lambda_r = 10^{-3}$, $h_x = 10^{-3}$, and for $\Psi^r$ we use the one described in App. G.2 with mean as the aggregate function.
- **SVI**. We use $K = 50$ to estimate the objective (Yin and Zhou, 2018, see Eq. (5)) which are around the values used in Yin and Zhou (2018). The implicit distribution is set to $r = \mathcal{N}(0, I_{d_z})$.
- **UVI**. For the HMC sampler, we follow in (Titsias and Ruiz, 2019) and use $50$ burn-in steps, with step-size $10^{-1}$ and $5$ leap-frog steps. The implicit distribution is set to $r = \mathcal{N}(0, I_{d_z})$.
- **SM**. For the "dual" function written as $f$ in the original paper (Yu and Zhang, 2023, see Algorithm 1) we use $\text{NN}(d_x, 512, d_x)$ and trained with RMSProp with stepsize $10^{-2}$. We tried a decaying learning schedule to $10-4$ but found that this degraded the performance. We used ReLU activations instead as we found that using leaky ReLUs harmed performance. The implicit distribution is set to $r = \mathcal{N}(0, I_{d_z})$.

**Computation Time.** For each run in the Concrete dataset with JIT compilation, PVI took  37 seconds, UVI took approximately 1 minute 40 seconds, SVI took  30 seconds, and SM took  27 seconds.

