# OpenReview forum: "Particle Semi-Implicit Variational Inference"
_NeurIPS.cc/2024/Conference — NeurIPS 2024 spotlight_

### Official Review · Reviewer_KYHY · 2024-07-09

**Soundness:** 4
**Presentation:** 3
**Contribution:** 3
**Rating:** 7
**Confidence:** 3

**Summary:**

This paper introduces Particle Variational Inference (PVI), a novel method for semi-implicit variational inference (SIVI) that employs empirical measures to optimize the mixing distribution without making parametric assumptions. Unlike existing SIVI methods that face challenges with intractable variational densities and rely on costly computational techniques, PVI directly optimizes the evidence lower bound (ELBO) using a particle approximation of an Euclidean–Wasserstein gradient flow.

**Strengths:**

1. The paper is well-organized and presents the complex concepts of the proposed method with clarity.

2. The Particle Variational Inference (PVI) algorithm, which discretizes the Euclidean–Wasserstein gradient flow to accommodate general mixing distributions, is both practical and innovative. This algorithm is not only grounded in solid theoretical foundations but also implemented effectively, showcasing its real-world applicability.

3. Empirical comparisons of PVI with other SIVI approaches across a variety of toy and real-world experiments demonstrate their good performance.

**Weaknesses:**

I do not identify any significant weaknesses in the paper. However, there are a few points that require clarification, which I have addressed in the questions section.

**Questions:**

1.	Figure 1: The current caption makes it difficult to interpret the figure. It would be helpful to explicitly mention that lighter shades represent smaller μ\muμ values, while darker shades represent larger μ\muμ values.
2.	Figure 2: The second sentence of the caption is somewhat confusing. Referring to Table 1 for additional clarification might improve the reader's understanding.
3.	Minor Typo: In line 303, "mu=2" should be corrected to "$\mu=2$"

**Limitations:**

The authors have adequately discussed the limitations of their approach.

---

> ### Author Rebuttal · Authors · 2024-08-06
>
> Thank you for your efforts in reviewing our work and for the helpful feedback and suggestions. All typographic errors will be fixed.
>
> > Figure 1: The current caption makes it difficult to interpret the figure. It would be helpful to explicitly mention that lighter shades represent smaller μ\muμ values, while darker shades represent larger μ\muμ values.
>
> This is a good suggestion. We shall make the necessary amendments.
>
> > Figure 2: The second sentence of the caption is somewhat confusing. Referring to Table 1 for additional clarification might improve the reader's understanding.
>
> Thank you for pointing out this mistake. The comment referred to the notation $\mu_{\pm \sigma}$ where $\mu$ is the average and $\sigma$ is the standard deviation computed from $10$ independent trials. This shall be fixed in future iterations.

---

> > ### Comment · Reviewer_KYHY · 2024-08-12
> >
> > Thank you for your response. I will keep my current score unchanged.

---

### Official Review · Reviewer_JYYZ · 2024-07-11

**Soundness:** 4
**Presentation:** 4
**Contribution:** 4
**Rating:** 9
**Confidence:** 5

**Summary:**

The authors introduce a novel algorithmic approach to fitting semi-implicit variational approximations. This method is based on discretizing a suitable gradient flow, and the paper provides a comprehensive theoretical analysis to support it. The approach, called Particle Variational Inference (PVI), directly optimizes over the space of probability distributions for the mixing distribution rather than optimizing a specific parametric form for the mixing distribution, as in previous SIVI approaches.

Numerical examples demonstrate improvements compared to existing semi-implicit variational inference methods. The paper provides a solid theoretical foundation and a practical algorithm with beneficial properties over existing methods in the literature.

**Strengths:**

This paper is a very strong submission, and I believe it is worthy of acceptance at this conference. The strengths that I would like to highlight are the following:

- **High degree of novelty:** Semi-implicit variational inference (SIVI) methods have existed for some time. Existing approaches faced challenges due to the nature of their design, often leading to optimizing bounds on the ELBO or difficult algorithms. This paper's essential contribution is constructing an objective function that can be directly optimized by its gradient flow. This is a significant advancement in the SIVI literature, enabling direct optimization of the ELBO.
- **Theoretical foundations:** The paper provides a solid theoretical analysis of the proposed method. This includes the study of a related gradient flow, establishing existence and uniqueness of solutions, and providing propagation of chaos results. These theoretical underpinnings give a rigorous basis for the practical algorithm.
- **Practical algorithm:** The particle variational inference (PVI) algorithm is derived as a practical implementation by discretizing the gradient flow of the proposed objective function. This direct link between theory and practice is a strength of the paper, as it provides a clear path from the mathematical formulation to a computationally feasible method.

**Weaknesses:**

The following points, particularly the limited experimental validation, prevent me from giving the submission a higher rating (8/9):
- **Limited scope of numerical experiments:** The paper's major weakness lies in its experimental section, which represents the minimum set of experiments for an acceptable paper. Specifically: a. Lack of empirical demonstration of expressiveness: Despite claiming in Section 2 that PVI can learn potentially more expressive variational approximations than other SIVI methods, no experiments empirically demonstrate this advantage. b. Insufficient exploration of optimization stability: The paper misses an opportunity to address a key challenge in SIVI methods - the difficulty in getting objective functions/algorithms to converge (or meaningfully work). Experiments showing that PVI provides consistently more stable optimization than other SIVI methods or converges on complex models where other SIVI methods struggle would have significantly strengthened the paper. Additionally, comparing performance accuracy on Bayesian Neural Networks (BNNs) for different SIVI methods doesn't show meaningful advantages to this method.
- **Theoretical-practical gap:** The theoretical analysis is for a modified gradient flow (with γ > 0), but the practical algorithm uses γ = 0. However, this discrepancy is mitigated by the authors' justification that this approach is relatively common in the literature, and empirical results did not show obvious discrepancies between theory and practice.
- **Clarity in explaining the flexibility of PVI's variational approximation:** The novelty in PVI's approach - not optimizing a specific parametric form of the mixing distribution - was not immediately apparent to me from the explanations in Sections 2 and 3 upon first read. In particular, the last sentence in Section 2 (lines 103-105) is confusing. While it becomes more understandable after reading Section 3, the intuition behind why Q_{YuZ} might not reduce to other parameterizations when fit with PVI is not fully explained.

**Questions:**

1. How sensitive is PVI to the choice of kernel and number of particles?
2. Can the theoretical analysis be extended to cover the γ = 0 case used in practice?
3. How does PVI scale to higher-dimensional problems or larger datasets?
4. Are there specific types of problems where PVI is expected to significantly outperform existing methods?
5. How does the choice of preconditioner affect PVI's performance?

**Limitations:**

The authors have adequately addressed limitations.

---

> ### Author Rebuttal · Authors · 2024-08-07
>
> Thank you for your hard work on our submission, and for recognizing the value of our work.
>
> > How sensitive is PVI to the choice of kernel and number of particles?
>
> We found that PVI the choice of kernel is an important one. In Section 3, we discuss the implications of the kernel choice more explicitly.
>
> As for the number of particles, after a certain number, we found that there were diminishing/no returns. In all our experiments, we used $100$ particles and found that this was sufficient for good performance. We did not finetune this quantity.
>
> > Can the theoretical analysis be extended to cover the γ = 0 case used in practice?
>
> Unfortunately, we do not currently know how to do this. Taking $\gamma \rightarrow 0$ will result in a non-Lipschitz drift which would violate the assumptions of our theoretical analysis. The bounds in Appendix F.4 would diverge.  In order to extend our analysis, one would need to establish existence and uniqueness via other arguments. However, the theory of McKean-Vlasov SDEs is at this stage much less developed than that for simple SDEs and we are not aware of any existing arguments which apply (if any) for the $\gamma = 0$. Our belief is that this reflects the current state of knowledge about McKean-Vlasov SDEs rather than being a fundamental issue.
>
> >  Despite claiming in Section 2 that PVI can learn potentially more expressive variational approximations than other SIVI methods, no experiments empirically demonstrate this advantage.
>
> In our experiments, we found that PVI performed well and outperformed all other existing semi-implicit VI methods. The advantage of our expressivity results in obtaining better approximations. In Section 5.2, the quality of the approximation can be seen through lower sliced Wasserstein scores and lower rejection rates; and in Section 5.3, we use MSE as a proxy of posterior quality for which PVI achieves the best overall performance. We also provide additional experiments as part of the rebuttal that compares MCMC samples with PVI samples on a Bayesian Logistic Regression example studied in [1, Section 5.4]. Although this does experiment does not demonstrate an advantage, it is encouraging that our method aligns with MCMC samples.
>
> [1] Yin, Mingzhang, and Mingyuan Zhou. "Semi-implicit variational inference." International conference on machine learning. PMLR, 2018.
>
> >  Are there specific types of problems where PVI is expected to significantly outperform existing methods?
>
> Whenever it's difficult to specify a variational family which is sufficiently expressive to capture the structure of the posterior even in this semi-implicit setting (e.g. in substantially multimodal settings) we would expect this approach to come into its own. In our numerical examples, we have focussed on fairly comparing the method with alternatives in settings in which they (other semi-implicit algorithms) do perform well (rather than contriving settings in which they fail) and have seen that the particle-based approach is extremely competitive even on the examples used to showcase earlier methods: we felt that demonstrating that nothing is lost in using this more general framework when other methods work provided a strong motivation for using it.
>
> However, we do not have an example where other methods will fail terribly (we have not actively tried to find one as we had focussed on comparing our method with early approaches using examples chosen by those authors to showcase their work; we can certainly explore this further and would welcome suggestions). Due to the mode-seeking behaviour of reverse KL that other methods minimize (at least approximately), we expect that these methods would (at worst) recover one of the modes for well-defined models. Please do share if you have any suggestions on this and we will include it in any future versions of the manuscript.
>
> > How does the choice of preconditioner affect PVI's performance?
>
> When a problem/kernel is ill-conditioned, we found that the preconditioner can be used to stabilize the training procedure. In the Bayesian neural network experiment (Section 5.3), we utilise this trick. In situations where the algorithm is already stable, it may not be required and the algorithm performed well regardless.
>
> > Clarity in explaining the flexibility of PVI's variational approximation.
>
> Thank you for pointing this out. We shall amend the final paragraph of section 2 to make this distinction clearer and more explicit.

---

> > ### Comment · Reviewer_JYYZ · 2024-08-13
> > **Response to rebuttal from authors**
> >
> > Thank you for providing the additional numerical results and answering my questions.
> >
> > The paper is technically flawless and is highly novel. Semi-implicit VI has been around for years, and the construction of the objective plus algorithm solves what I consider an existing open problem in the area (unbiased gradient estimator of the ELBO directly).
> >
> > Thanks for clarifying the existing numerical results. After an additional read, I am super satisfied with the benchmark versus the existing methods provided, and the wide variety of "metrics" used for assessment. While none of the target densities are in the "wow factor" territory, this is not necessary in my opinion. The paper is so strong, that I am sure that the focus of derivative works could be to "apply" or adapt the existing algorithm for high-dimensional and/or more difficult targets.
> >
> > Given that VI is the most popular approximate inference method at the moment, and that such approaches are heavily related to algorithms used to train quite a few of the popular classes of generative models at the moment, this paper has room to be of interest to many in the community, and I would some sort of spotlight or oral for this work. Thanks for the great submission.

---

### Official Review · Reviewer_38kT · 2024-07-12

**Soundness:** 3
**Presentation:** 3
**Contribution:** 3
**Rating:** 7
**Confidence:** 2

**Summary:**

The authors propose a method for SIVI called Particle Variational Inference (PVI) which employs a particle approximation of an Euclidean–Wasserstein gradient flow.  PVI directly optimizes the ELBO, and it makes no parametric assumption about the mixing distribution. Their empirical results demonstrate that PVI performs favorably against other SIVI methods across various tasks. The authors provide a theoretical analysis of the behavior of the gradient flow of a related free energy functional.

**Strengths:**

The authors provide extensive theoretical analysis to support the proposed method. The authors provide solid theoretical results along with the proposed method. The paper is well written.

**Weaknesses:**

Experiments on a real-world application could strengthen the paper.

**Questions:**

Line 329: what's the meaning of 'Here the ± denotes the average ...'?

---

> ### Author Rebuttal · Authors · 2024-08-06
>
> Thank you for your time spent reviewing our work.
>
> > what's the meaning of 'Here the ± denotes the average ...'?
>
> Thank you for pointing out this mistake. The comment referred to the notation $\mu_{\pm \sigma}$ where $\mu$ is the average and $\sigma$ is the standard deviation computed from $10$ independent trials. This shall be fixed in future iterations.

---

> > ### Comment · Reviewer_38kT · 2024-08-11
> >
> > Thanks for the response. The score from this reviewer is unchanged.

---

### Official Review · Reviewer_o93p · 2024-07-13

**Soundness:** 3
**Presentation:** 2
**Contribution:** 3
**Rating:** 6
**Confidence:** 4

**Summary:**

The paper proposes PVI as a new method to conduct variational inference using the semi-implicit distribution. The method is to construct a gradient flow to minimize a regularized ELBO, which is practically implemented as the particle propagations. The empirical studies show the accuracy over density estimation and posterior predictions.

**Strengths:**

- The idea of using Wasserstein gradient flow to optimize the intractable ELBO of SIVI is novel.
- The method has good accuracy in the provided simulations.
- The techniques in the method derivation may be useful for other areas such as flows and diffusion models.

**Weaknesses:**

A major concern is about the simulations. First, it is unclear whether the comparisons between PVI and the baselines SVI, UVI and SM are fair. The accuracy of SIVI methods can often improve with an increase in the computation such as the number of samples in SVI and the MCMC steps in UVI. For a fair comparison, all methods need to be under the same computation budget. However, the current paper does not provide details of how the SVI, UVI, and SM are implemented. It would be better to include figures that show computation/time versus the accuracies for all the methods.

Current simulations are relatively too simple and are only conducted on several 2D toy examples and UCI datasets. First, none of these simulations directly show the accuracy of the posterior inference for a complete Bayesian model. Second, all the simulations are in low-dimension. Does the particle method suffer from the curse of dimension problem? How does the number of particles scale with dimensionality to maintain accuracy?  It would be interesting to verify uncertainty estimation in high dimensions.

Last, the paper writing is not coherent in some places. For example
- The non-coercive is not defined in Proposition 2
- The paper mentions the non-coercive "is closely related to the problem of posterior collapse"; what is the relationship exactly?
- How to compute the precondition matrices in Eq 13 is not discussed

**Questions:**

Does the regularization in Eq 4 introduce bias in the posterior inference?

Does the theoretical analysis in Sec. 4 provide guidance in designing the algorithm?

How to check coercive and l.s.c. in Proposition 3?

---

> ### Author Rebuttal · Authors · 2024-08-07
>
> Thank you for the effort you spent on our work.
>
> > However, the current paper does not provide details of how the SVI, UVI, and SM are implemented.
>
> We perhaps didn't make this clear enough (and can address that in subsequent versions), but line 281 of the main text points out that all hyperparameter choices and runtimes for all methods are provided in Appendix H. We usually follow the recommended settings found for each competing method in their respective papers (except for when we did not find explicit recommendations). Source code for our implementations of all methods was provided at the time of submission.
>
> There is a question of whether runtime is a meaningful measure of fairness given that there can be discrepancies based on how an algorithm is implemented. As the runtimes provided in Appendix H demonstrate, other algorithms were not disadvantaged. For instance, we followed the recommendations of UVI paper which resulted in an algorithm that exceeded our time computation by an order of magnitude. In the supplementary PDF, we include the requested figures for the density estimation task. To achieve high speeds with PVI, one takes advantage of the fact that the particle update is easily parallelizable.
>
> > First, none of these simulations directly show the accuracy of the posterior inference for a complete Bayesian model.
>
> We agree. We now provide (preliminary) experiments on the Bayesian Logistic Regressions setup studied in prior works [1, Section 5.4]. In the attached PDF, we provide plots that compare the quality of the posterior against MCMC samples obtained in [1]. It can be seen that PVI obtains a posterior quality that closely matches that of the MCMC samples.
>
> [1] Yin, Mingzhang, and Mingyuan Zhou. "Semi-implicit variational inference." International conference on machine learning. PMLR, 2018.
>
> > Does the particle method suffer from the curse of dimension problem? How does the number of particles scale with dimensionality to maintain accuracy?
> >all the simulations are in low-dimension.
>
> As noted in Appendix 5.3, the Bayesian neural network example has dimensionality {331, 81, 101} which one can claim is not "low dimensional". Like other methods, PVI utilizes neural networks to allow the method to scale to high dimensions while retaining the expressivity of the particle method.
> Experimentally, we found that $100$ particles were sufficient for good performance across the examples considered. We kept this constant throughout our experiments and did not finetune it. The source code for our implementations was provided at the time of submission.
>
> >Last, the paper writing is not coherent in some places
> The non-coercive is not defined in Proposition 2.
>
> The coercivity definition is a standard one for which we provided an explicit reference for it in the main text on line 128. We shall add the definition to the Appendix.
>
> We will reread the manuscript carefully and address any similar issues; we'd be happy to address any specific concerns you may still have.
>
> > The paper mentions the non-coercive "is closely related to the problem of posterior collapse"; what is the relationship exactly?
>
> In the amortized VI, one definition of posterior collapse is when the approximate posterior $q(x|z)$ does not depend on $z$ and collapses to the prior. In the context of PVI, one can prove that the functional $\cal{E}$ is not coercive by using the kernel $k(x|z)$ that does not depend on $z$. Since the kernel is $k$ and approximate posterior $q$ often share the same functional form, e.g., they ($k$ and $q$) are often parameterized with $\cal{N}(\mu(z), \sigma^2I)$ where $\mu$ is some neural network, the comment suggests to the reader why it may be unsurprising if they share this issue.
>
> > How to compute the precondition matrices in Eq 13 is not discussed
>
> The preconditioning matrix was discussed in Appendix G.2 for which we have provided a reference at line 209 which is within the main text.
>
> > Does the regularization in Eq 4 introduce bias in the posterior inference?
>
> Yes. As with any non-parametric estimation problem with a finite sample size, regularization is an important algorithmic step. Loosely speaking, one can decrease the weighting of the regularization as the number of samples goes towards infinity to remain consistent.
>
> > Does the theoretical analysis in Sec. 4 provide guidance in designing the algorithm?
>
> Not directly. The primary purpose of the analysis is to provide "theoretical underpinnings give a rigorous basis for the practical algorithm" to quote Rev JYYZ.
>
> > How to check coercive and l.s.c. in Proposition 3?
>
> There are various proof techniques that one can use to show coercivity and lsc. Our proposed regularizer satisfies the assumptions of Proposition 3.

---

> > ### Comment · Reviewer_o93p · 2024-08-10
> > **Thank you for the rebuttal**
> >
> > The further analysis of the computation time and full Bayesian model, together with the discussion on the dimensionality, strengthen the paper and address my major concerns. Thanks the authors for providing code to facilitate results reproducing. I update my score accordingly.

---

### Author Rebuttal · Authors · 2024-08-07

We thank all of the referees for their helpful and thoughtful comments; we were pleased that three of the four reviewers reacted positively to the initial submission and hope that we can address the points that were raised during the reviewing process here.

The main area of concern overall appears to have been the extent of the numerical evaluation of the algorithm, although referees differed slightly on the things that they would have liked to see beyond those experiments already provided. We have now added an additional example, a Bayesian Logistic Regression, thereby including most of the examples considered by previous work on semi-implicit variational inference and allowing us to show prior approximation quality on a full Bayesian model and compare with alternative Monte Carlo Markov Chain methods in this challenging setting. In addition, for our prior experiments in Section 5.2, we explore runtime behaviour in further detail.

Aside from numerical evaluation:
* One referee found some lack of clarity in the exposition: we hope this is now addressed.
* All specific points of detail raised have been addressed.
* All other identified weaknesses and questions have been responded to below: we are not able to close the theoretical-practical gap, but this is common to many methods based around the nascent field of mean field approximation of McKean-Vlasov SDEs rather than being specific to our work; we have otherwise been able to at least partially answer all of these.

If you have any further questions or feel that we have not adequately addressed any of the points which were raised, then please let us know in the discussion and we shall endeavour to do so.

---

### Decision · Program_Chairs · 2024-09-25

**Decision:**

Accept (spotlight)

**Comment:**

This paper proposes an algorithm called Particle Variational Inference (PVI) to fit semi-implicit variational distributions using gradient flows. Unlike previous SIVI approaches, PVI directly optimizes over the space of probability distributions for the mixing distribution. The paper provides solid theorical contributions, as well as numerous experiments, using multiple metrics and settings.

All four reviewers support acceptance of the paper. Reviewer JYYZ mentions the paper could potentially influence the area of generative modeling, as the proposed approximate inference tools could be used for training generative models.

However, one main weakness, shared by several reviewers, is the dimensionality of the considered problems (which is moderate but not high), which indicates that future research may be needed to target higher-dimensional distributions.

Given the novelty and good execution of the paper, I believe this is not necessarily a limitation, and recommend acceptance.